



# Classifying aerosol type using in situ surface spectral aerosol optical properties

Lauren Schmeisser[*][1,2], Elisabeth Andrews[1,2], John A. Ogren[1], Patrick Sheridan[1], Anne Jefferson[1,2], Sangeeta Sharma[3], Jeong Eun Kim[4], James P. Sherman[5], Mar Sorribas[6], Ivo Kalapov[7], Todor Arsov[7], Christo Angelov[7], Olga L. Mayol-Bracero[8], Casper Labuschagne[9,10], Sang-Woo Kim[11], András Hoffer[12], Neng-Huei Lin[13], Hao-Ping Chia[13], Michael Bergin[14], Junying Sun[15], Peng Liu[16], Hao Wu[16]

[1]National Oceanic and Atmospheric Administration, Earth Systems Research Laboratory, Boulder, CO, USA
[2]University of Colorado at Boulder, CIRES, Boulder, CO, USA
[3]Environment and Climate Change Canada, Science and Technology Branch, Ontario, Canada
[4]Environmental Meteorology Research Division, National Institute of Meteorological Sciences
[5]Appalachain State University, Boone, NC, USA
[6]Atmospheric Sounding Station-El Arenosillo, Atmospheric Research and Instrumentation Branch, INTA, 21130, Mazagón, Huelva, Spain
[7]Institute for Nuclear Research and Nuclear Energy by the Bulgarian Academy of Sciences, Sofia, Bulgaria
[8]University of Puerto Rico, Department of Environmental Science, San Juan, PR, USA
[9]South African Weather Service, Stellenbosch, South Africa
[10]Unit for Environmental Sciences and Management, North-West University, Potchefstroom Campus, South Africa
[11]Seoul National University, Seoul 08826, Korea
[12]MTA-PE Air Chemistry Research Group, Veszprém, P.O. Box 158, H-8201, Hungary
[13]National Central University, Department of Atmospheric Sciences, Chung-LI, Taoyuan City, Taiwan
[14]Duke University, Department of Civil & Environmental Engineering, Durham, NC
[15]State Key Laboratory of Severe Weather & Key Laboratory of Atmospheric Chemistry of CMA, Chinese Academy of Meteorological Sciences, Beijing 100081, China
[16]China GAW Baseline Observatory, Qinghai Meteorological Bureau, Xining 810001, China
*Correspondence to*: Lauren Schmeisser (lauren.schmeisser@gmail.com)

*Corresponding author now at: University of Washington, Department of Atmospheric Sciences, Seattle, WA, USA

**Abstract**

Knowledge of aerosol size and composition is important for determining radiative forcing effects of aerosols, identifying aerosol sources, and improving aerosol satellite retrieval algorithms. The ability to extrapolate aerosol size and composition, or type, from intensive aerosol optical properties can help expand the current knowledge of spatio-temporal variability of aerosol type globally, particularly where chemical composition measurements do not exist concurrently with optical property measurements. This study uses medians of scattering Ångström exponent (SAE), absorption Ångström exponent (AAE) and single scattering albedo (SSA) from 24 stations within the NOAA federated aerosol network to infer aerosol type using previously published aerosol classification schemes.

Three methods are implemented to obtain a best estimate of dominant aerosol type at each station using aerosol optical properties. The first method plots station medians into an AAE vs. SAE plot space, so that a unique combination of intensive properties corresponds with an aerosol type. The second typing method expands on the first by introducing a multivariate cluster analysis, which aims to group stations with similar optical characteristics, and thus similar dominant aerosol type. The third and final classification method pairs 3-day backward air mass trajectories with median aerosol optical properties to explore the relationship between trajectory origin (proxy for likely aerosol type) and aerosol intensive parameters, while allowing for multiple dominant aerosol types at each station.

The three aerosol classification methods have some common, and thus robust, results. In general, estimating dominant aerosol type using optical properties is best suited for site locations with a stable and homogenous aerosol population,



particularly continental polluted (carbonaceous aerosol), marine polluted (carbonaceous aerosol mixed with sea salt), and continental dust/biomass sites (dust and carbonaceous aerosol); however, current classification schemes perform poorly when predicting dominant aerosol type at remote marine and Arctic sites, and at stations with more complex locations and topography where variable aerosol populations are not well represented by median optical properties.

Although the aerosol classification methods presented here provide new ways to reduce ambiguity in typing schemes, there is more work needed to find aerosol typing methods that are useful for a larger range of geographic locations and aerosol populations.

## 1. Introduction

Although it is well established that aerosol particles affect the radiative forcing of climate both directly by scattering

and absorbing sunlight and indirectly by influencing cloud formation and precipitation, aerosols still remain a primary source of uncertainty in assessing the Earth's radiative budget (Boucher et al., 2013). This uncertainty arises from a large range of aerosol chemical and physical properties, as well as from the high spatio-temporal variability of aerosol particles. In order to help reduce this uncertainty and be able to better predict climatic effects of aerosols, there is a need for long-term global monitoring of aerosols (Hansen et al., 1996), compiling records not only of aerosol loading but

also of aerosol characteristics and type.

Determination of aerosol type (e.g., black carbon, sea salt, dust, etc.), which is defined by the size and composition of an aerosol, is important in characterizing the role of aerosols in atmospheric processes and feedbacks, since different aerosol types have different radiative forcing effects and atmospheric behavior. Additionally, knowledge of aerosol type helps identify aerosol source, which can be useful in implementing controls or policies to reduce aerosols that

negatively influence air quality and public health, and also to better understand atmospheric dynamics and long-range transport. Constraining aerosol type is also needed for improving aerosol satellite retrieval algorithms and for validating climate models (Russell et al., 2014).

Recent studies, discussed below, present classification schemes to infer aerosol type from intensive optical properties, which are calculated from ratios of extensive properties and thus not directly dependent on the aerosol amount.

Successful application of this method could allow for access to aerosol composition information from remote or in situ optical property measurements that do not otherwise provide an indication of aerosol type.

## 2. Background

Three optical properties that hold information on aerosol type include scattering Ångström exponent (SAE), absorption Ångström exponent (AAE) and single scattering albedo (SSA). SAE represents the wavelength dependence of

scattering, and varies inversely with particle size, so that small values of SAE indicate larger aerosol particles (e.g., dust and sea salt), and large values of SAE indicate relatively smaller aerosol particles (Schuster et al., 2006; Bergin et al., 2000 and references therein). AAE represents the wavelength dependence of absorption and depends on the composition of absorbing aerosols, such that aerosol materials have a unique range of AAE values (Russell et al., 2010; Bergstrom et al., 2002, 2007). Black carbon (BC), for example, has a theoretical AAE value of 1, while dust aerosol

typically has AAE values greater than 2 (Bergstrom et al., 2002, 2007; Kirchstetter et al., 2004). SSA is the ratio of scattering to extinction (absorption + scattering) and provides information on aerosol darkness and composition, and may determine the net sign of an aerosol's radiative forcing (Hansen et al., 1997). High SSA values near 1 indicate low- or non-absorbing "white" aerosols, while low SSA values (below 0.85) indicate "darker" highly-absorbing aerosols, and





thus an SSA value can be used to characterize the aerosol type (Bergstrom et al., 2002; Russell et al., 2010; Gyawali et al., 2012). Equations for calculating these properties from extensive optical parameters are found in the Methods section. Many studies have used the information inherent in these optical properties to predict aerosol type; Table 1 provides a review of previous studies that have utilized intensive optical property thresholds to identify aerosol type.

The studies listed in Table 1 all take slightly different approaches to show that intensive aerosol optical properties (SAE, AAE, and SSA) can be utilized to classify aerosol type. Bahadur et al. (2012) determine a scheme to partition various absorbing aerosol types based on absorbing aerosol optical depth measurements from numerous AERONET sites that represent a single absorbing aerosol, and test the proposed scheme using California AERONET sites with mixed aerosols. Cazorla et al. (2013) also make use of California AERONET sites by combining the measured aerosol optical
properties with in situ aerosol chemical composition measurements from an aircraft campaign to create a matrix that delineates aerosol type in an AAE vs. SAE plot space. Eleven AERONET sites from around the globe are used in the study by Russell et al. (2010) to show that AAE values from full column measurements are highly correlated with aerosol type, in general agreement with the two previously mentioned AERONET aerosol typing schemes that suggest AAE values near 1 indicate fossil fuel burning aerosol, higher AAE values indicate OC/biomass burning aerosols, and
the highest AAE values indicate dust aerosols.

In situ measurements have also been used for aerosol classification schemes. In situ optical measurements from the INTEX-NA aircraft campaign are used by Clark et al. (2007) to separate biomass burning from pollution plumes. Costabile et al. (2013) propose a scheme to classify aerosols based on absorption and scattering values, using 2 years of in situ urban data from Rome, Italy coupled with numerical simulations to create a paradigm linking key aerosol
populations to their unique aerosol optical properties. Six months of optical property measurements from the in situ monitoring site in Gosan, South Korea are used by Lee et al. (2012) and categorized by air mass type (either pollution or dust) using chemical composition, back trajectories and meteorological conditions, and SAE and AAE values are analyzed, yielding results that show dust air masses have the highest AAE values, with organic carbon (OC) polluted air masses showing the next highest AAE values. Cappa et al. (2016) utilized surface in situ measurements from the
CARES field campaign to categorize aerosol they observed and to suggest some modifications to the Cazorla et al. (2013) aerosol classification scheme. Finally, Yang et al. (2009) used the distinct SSA, AAE and SAE values of different air plumes in the EAST-AIRE campaign to identify absorption contributions from desert dust, biomass burning, industrial plumes, and clean air in Beijing, China. It is worth mentioning that some studies take into account the spectral dependence of SSA in aerosol classification schemes (Li et al, 2015; Russell et al., 2010). This parameter
was calculated for the monitoring stations in this study, but was not useful in classifying aerosol type compared to the other optical properties discussed; therefore, the spectral dependence of SSA is not discussed in this study.

Care must be taken in comparing thresholds from all aforementioned studies, as differences are likely between column-average, ambient AERONET measurements and low-RH, surface in situ measurements. Furthermore, different wavelength pairs are used to calculate AAE and SAE depending on the study. In general, however, all studies suggest
that AAE values around 1 represent BC and/or fossil fuel burning aerosols, higher AAE values indicate OC and/or dust and the highest AAE values indicate brown carbon, and that high SAE values are associated with small anthropogenic aerosols (e.g., BC, sulfates, or nitrates) and low SAE values are associated with large aerosols like sea salt and dust.

This paper aims to assess the applicability of previous typing methods/schemes to data from 24 in situ monitoring sites within the NOAA/ESRL Federated Monitoring Network, and to explore how typing schemes may be improved based





on methods using cluster analyses and air mass back trajectories. The following questions are addressed: (1) Are the relationships between SAE and AAE data from 24 stations in the NOAA federated monitoring network consistent with relationships used to identify dominant aerosol type using aerosol classification schemes previously reported in the literature?; (2) Can multivariate cluster analyses on aerosol properties be used to reduce both the ambiguity in inferring

likely dominant aerosol type from median aerosol optical properties, and the uncertainty in aerosol type optical property thresholds?; and (3) How can back trajectory clusters and subsequent information on air mass source help elucidate multiple aerosol types at individual sites?

The literature on classifying aerosols has been largely dominated by analysis of ground-based remote sensing or satellite data (Cazorla et al., 2013; Russell et al., 2010; Russell et al., 2014; Omar et al., 2005; Giles et al., 2012;

Bergstrom et al., 2007; Bergstrom et al., 2010; Bahadur et al., 2012; Dubovik, 2002), with fewer analyses done using surface in situ aerosol optical property measurements (Cappa et al., 2016; Costabile et al., 2013; Yang et al., 2009; Lee et al., 2012). The analyses in this paper utilize ground-based in situ spectral optical data that afford a unique insight into long-term, quality-assured point observations. Furthermore, since the in situ data sets used in this study are not restricted by AOD thresholds as are AERONET data sets, they offer a more thorough look at regions with relatively

clean air.

Unlike most previous studies, this study looks at long-term records of aerosol optical properties, and does so at a wide range of geographic locations, including mountaintop, desert, continental and coastal sites. Not only does the study offer a wide range of aerosol types to be analyzed in an individual geographic location, but provides analysis of the same aerosol type in different geographic locations.

## 3. Site descriptions

This study investigates aerosol populations at 24 monitoring stations in the NOAA/ESRL Federated Aerosol Monitoring Network. Sites were selected for the study based on availability of data - each site had to meet the following criteria: (1) aerosol optical data available at 3 wavelengths, and (2) long-term (>6 months) continuous measurement records of scattering and absorption coefficients during the two year time period 2012-2013, unless otherwise noted (see Table 2

for time range for each site). The ARM Mobile Facility (AMF) (part of the U.S. Department of Energy's ARM Climate Research Facility) deployments, indicated in bold in Table 2, are typically one to two year deployments. Most of the AMF measurement times do not overlap with the 2012-2013 analysis period, but should nevertheless be comparable to other sites, and are included as a means of broadening the range of geographic locations for the analysis. One advantage of this study is the wide diversity of location types and observed aerosol loadings (which span over 3 orders of

magnitude). This study includes sites in both the northern and southern hemispheres, ranging in altitude from sea level to 3800 m above sea level (asl), and with various climate regimes including marine, continental and Arctic. The sites experience different levels of anthropogenic influence ranging from clean to very polluted. The 24 stations are described in Table 2, and Fig. 1 shows a map of the stations.

Table 2 presents monitoring site location, latitude, longitude, altitude, scattering and absorption instruments, size cuts,

date range of data utilized, site classification, and site description for 24 monitoring stations in the NOAA/ESRL Federated Aerosol Monitoring Network. Bolded station names in the table indicate sites where the short term AMF was deployed.

Sites are categorized based on the site's geography and surrounding land use. Arctic sites are at latitudes greater than





70 N. Continental polluted sites have influence from urban and industrial pollution. Continental dust/biomass sites are generally more rural with influence from desert dust and/or biomass burning. Marine clean sites are in remote coastal locations, have little influence from pollution sources (except perhaps from long-range transport events), and see an abundance of marine aerosols. Marine polluted sites are also in coastal locations and may measure pollution aerosols

(from continental air masses) or marine aerosols (from oceanic air masses) or some combination thereof, depending on the wind direction. Mountaintop classifications indicate sites that are higher than 2800m in elevation; these high altitude monitoring stations sample both free troposphere air and air masses transported from lower elevations due to upslope/downslope flow. Site classification is inherently subjective and not always clear-cut. The authors acknowledge that sites could be considered to have more than one classification and have multiple aerosol types. However, the

classifications were designated based on 'best fit' to the site characteristics and are representative of the dominant aerosol type at each site.

### 4. Data and Instruments

The data sets used for the analysis are comprised of in situ scattering and absorption coefficients ($\sigma_{sp}$ and $\sigma_{ap}$, respectively), which are quality assured and used to calculate additional parameters (AAE, SAE and SSA) as described

in Eqs. (1)-(3). One-hour averaged data are used for the assessment of aerosol classification schemes and the multivariate cluster analysis. However, we use 6-hour averaged optical properties for the back trajectory analysis, since back trajectories are run at 6-hour intervals. Datasets from NOAA and collaborators are publically available from the World Data Center for Aerosols (http://ebas.nilu.no/), with the exception of WLG data, while the AMF datasets are publically available from DOE (http://www.arm.gov/).

Scattering coefficients were obtained with a TSI 3563 integrating nephelometer (TSI Inc.) at all sites, operating at wavelength channels 450, 500 and 700 nm. Absorption coefficients were measured by either a 3-wavelength Particle Soot Absorption Photometer (PSAP, Radiance Research), or a 3-wavelength Continuous Light Absorption Photometer (CLAP, NOAA). The PSAP instruments operate at wavelengths 467, 530, and 660 nm, and CLAP instruments operate at wavelengths 467, 528, and 652 nm. In either case, the $\sigma_{ap}$ values are corrected to 450,550, and 700nm (using AAE) so

as to match the wavelengths of the $\sigma_{sp}$ measurements.

Table 2 indicates which instruments operate at each station. At MLO and BND, data from both the PSAP and CLAP were utilized, since at both stations the PSAP was replaced with a CLAP in the middle of the study period. A comparative analysis of PSAP and CLAP measurements shows that the two instruments produce comparable measurements, and thus combining or directly comparing data from both instruments is not expected to affect results

(Ogren et al., 2013).

To ensure datasets are comparable across monitoring stations, all data are quality controlled. In order to minimize aerosol hygroscopic effects, measurements at all stations are made at a reduced relative humidity (RH < 40%) by heating the inlet air or by diluting with filtered, dry air. Monitoring station buildings are also temperature controlled, and inlet stacks have protective caps and screens to prevent interference from precipitation, insects or debris. All aerosol

scattering coefficient measurements from the TSI nephelometers are corrected for angular non-idealities using corrections from Anderson and Ogren (1998). After the corrections, scattering coefficients measured by the nephelometer have an uncertainty of 9.3% for the 10 μm size cut, based on the analysis by Sherman et al. (2015). The Sherman et al. (2015) calculations represent median continental conditions, and might change at sites with cleaner or more polluted conditions. Aerosol absorption coefficient measurements from PSAP and CLAP instruments are adjusted





for flow rate, spot size, and aerosol scattering, using the correction from Bond et al. (1999) and further adjusted for wavelength based on corrections from Ogren (2010). After corrections, absorption coefficients measured by the PSAP or CLAP have an uncertainty of ~20% (Sherman et al., 2015). Finally, all data are passed through a quality assurance/quality control editing process in which measurement records are screened for atypical aerosol parameters

(see Delene and Ogren (2002) and Sheridan et al. (2015) for detailed descriptions of quality assurance and quality control procedures). Points that appear anomalous due to local pollution sources (non-representative of regional aerosol), instrument error or excessive noise are not included in this analysis.

The measured scattering and absorption coefficients are extensive aerosol properties because they depend on the amount of aerosol present (Ogren, 1995; Delene & Ogren, 2002). Intensive aerosol optical properties are calculated

from ratios of the extensive properties. The aerosol intensive properties including absorption Ångström exponent (AAE), scattering Ångström exponent (SAE), and single scattering albedo (SSA), are of primary interest to this study since they contain information on aerosol size or composition, and are calculated as indicated in the following equations:

$$AAE_{\lambda 1/\lambda 2} = \frac{-\log(\frac{\sigma_{ap,\lambda 1}}{\sigma_{ap,\lambda 2}})}{\log(\frac{\lambda_1}{\lambda_2})} \quad (1)$$

$$SAE_{\lambda 1/\lambda 2} = \frac{-\log(\frac{\sigma_{sp,\lambda 1}}{\sigma_{sp,\lambda 2}})}{\log(\frac{\lambda_1}{\lambda_2})} \quad (2)$$

$$SSA_{\lambda 1} = \frac{\sigma_{sp,\lambda 1}}{\sigma_{sp,\lambda 1} + \sigma_{ap,\lambda 1}} \quad (3)$$

where $\sigma_{ap,\lambda 1}$ represents absorption coefficient at wavelength $\lambda_1$, and $\sigma_{ap,\lambda 2}$ represents absorption coefficient at wavelength $\lambda_2$. Similarly, $\sigma_{sp,\lambda 1}$ and $\sigma_{sp,\lambda 2}$ represent scattering coefficients at wavelengths $\lambda_1$ and $\lambda_2$, respectively. Unless otherwise indicated, all data presented here refer to the green wavelength channel (550nm) for SSA, absorption, and

scattering coefficient values, or the blue/red wavelength pair (450nm/700nm) for the SAE and AAE values. CLAP and PSAP wavelengths were adjusted to match the nephelometer wavelengths to compute the intensive variables.

Only aerosol measurements where $\sigma_{sp} > 1$ Mm$^{-1}$ and $\sigma_{ap} > 0.5$ Mm$^{-1}$ are included in the analyses. Data below these values are less reliable due to instrument noise at low aerosol loading, thus the constraints are meant to act as noise thresholds. This inherently adds bias to the data, as monitoring sites with consistently low absorption and scattering

coefficients may end up with limited data points after the thresholds are applied, leaving measurement records with higher loadings that may not be fully representative of typical aerosol populations at the site. This constraint has the greatest effect on clean sites like ALT, BRW, and SUM (which measure Arctic air), BEO and MLO (which sometimes measure free tropospheric air), and CPR, CPT, PVC, PYE, and THD (which sometimes measure clean marine air). The constraints push the extensive scattering and absorption values higher. More details on the effect of the thresholds on

the analysis of clean stations can be found in Table S5 in the supplemental materials.

There are some differences in monitoring station data that may affect the results of the following analyses, and are noted here. SUM utilizes a 2.5 μm size cut, while all other stations use a size cut of 1 and 10 μm, but only the 10 μm data are used in this study. This size cut discrepancy will bias SUM data towards higher SAE values than would be





found with a larger size cut. Since ARM station data records are typically less than one year in length, while all other station data are 2 years in length, any site-specific seasonal variations may not be captured in the ARM data records. Furthermore, ARM measurement times and CPT times typically do not overlap with the baseline study period of 2012-2013, so any extreme events specific to those years are not reflected in the CPT (data only from years 2010-2011) or

ARM (FKB, GRW, NIM, PGH, PVC, PYE) sites measurements.

### 5. Results

#### 5.1 Application and assessment of previous aerosol typing schemes

Like many previous studies (Cappa et al., 2016; Cazorla et al., 2013; Costabile et al., 2013; Yang et al., 2009; Russell et al., 2010; Lee et al., 2012; Bahadur et al., 2012), an AAE vs. SAE plot space is used here to visualize relationships

between aerosol optical properties and likely aerosol type. Since SAE indicates aerosol size and AAE holds information on aerosol composition and size, a unique combination of the two, and thus where that combination falls within the AAE vs. SAE plot space, suggests a particular aerosol type. Most previous studies use chemical composition data (Costabile et al., 2013; Lee et al., 2012; Cazorla et al., 2013) or numerical simulations (Costabile et al., 2013) to validate the proposed aerosol classification scheme; however, since neither of those methods are available for this study,

thresholds from previous studies are used here to infer likely dominant aerosol type, and results are assessed based on knowledge of the site. For the first iteration of the analysis, long-term optical property medians from multiple stations are presented in one plot space, for a comparative overview of inferred dominant aerosol type at many sites.

The median and interquartile spread of SAE, AAE, SSA, scattering coefficient and absorption coefficient values at each site are presented in Table 3. Additionally, Table 3 indicates the aerosol type as determined by the Cazorla et al. (2013)

matrix overlaid on the plot of optical property medians in Fig. 2(b) ('aerosol type before clustering'), as well as the aerosol type determined from a clustering analysis ('aerosol type after clustering'), as described in the next section. Descriptions of the aerosol types can be found in Cazorla et al. (2013).

Median AAE and SAE values for each station are shown in Fig. 2(a) along with bars that represent the interquartile spread (25th to 75th percentiles) of the data. Points are shaded by median SSA value at that station. Medians are used in

order to minimize influence from outliers. There are no strong spatial patterns visible in SSA shading within the AAE vs. SAE plot space in Fig. 2(a). Stations with high median SAE (smaller particles) tend to have slightly lower median SSA values (darker particles) than those with low median SAE, and vice versa. However, there are exceptions to this tendency, with NIM having a low median SAE value and relatively low median SSA, and PVC having a high median SAE value and relatively high median SSA. Previous studies established that SSA and the wavelength dependence of

SSA can be used to signify aerosol type (Yang et al., 2009; Russell et al., 2010). A three-dimensional plot space helps visualize the relationships amongst SAE, AAE and SSA. This will be further explored in the next section.

Figure 2(a) shows the wide variance of intensive properties at any one site, with values spanning beyond the optical property signatures of a single aerosol type. For example, CPR has interquartile AAE values ranging from 1.16-2.65, a spread that encompasses multiple potential aerosol compositions, as outlined by the thresholds in Table 2 and by the

classification matrix in Fig. 2(b). Interquartile ranges conservatively bound the intensive properties and thus represent the dominant aerosol type at each monitoring site. Some, if not all, of the sites could have multiple aerosol types that are not well represented by the medians illustrated in Fig. 2, as discussed in the next section.





Figure 2(b) shows the same optical property medians that are plotted in Fig. 2(a). Station points are colored by station location type (as listed in Table 2), with the aerosol classification matrix from Cazorla et al. (2013) is overlaid on the plot space. Optical properties from the 24 NOAA Federated Network stations were evaluated with multiple existing published aerosol classification schemes; however, given the clear visualization and complete characterization of the

parameter space afforded by the Cazorla et al. (2013) matrix, that is the only scheme used for a visual comparison in this study. The station location type provides the reader guidance on what aerosol types might be expected at the site.

There is a natural clustering of all continental polluted sites on the right hand side of the plot in Fig. 2(b), in the section Cazorla et al. (2013) designated as EC/OC aerosol. Median AAE>1 at these sites is consistent with other studies (Russell et al., 2010; Lee et al., 2012; Yang et al., 2009; Cazorla et al., 2013). Furthermore, both remote/clean marine

(e.g., GRW, PYE, THD) sites and dust-influenced sites (e.g., NIM) tend to fall on the left hand side of the plot with low SAE values, indicative of sea salt, highly processed and coated particles, or dust (Cappa et al., 2016; Cazorla et al., 2013; Lee et al., 2012; Clarke et al., 2007; Yang et al., 2009). The largest median AAE values are observed at NIM and CPR, both of which experience Saharan dust events. NIM is located at the southern edge of the Saharan desert. Dust transport to CPR is predominantly from the African Sahel region (Propero et al., 2014). Although ARN experiences

Saharan dust events (Toledano et al., 2007), these events are not frequent enough to substantially influence the median in situ aerosol optical properties. The high AAE values at sites influenced by dust agree with the findings of Russell et al. (2010), Lee et al. (2012), and Yang et al. (2009), which identified dust aerosol as having the largest AAE values of observed aerosol types. These sites also fit in well with the Cazorla et al. (2013) matrix. Aerosol types assigned to the marine THD, ARN, GRW, PYE, CPT, CPR, and PVC sites by the Cazorla et al. (2013) aerosol classification scheme

exhibit high variance in their properties, indicating a diverse influence of aerosol. For example, the high SAE values at PVC show the strong influence of transport from nearby urban centers of Boston and Providence as well as pollution from summer traffic on Cape Cod, which dominate the effect of marine aerosol on the site's median SAE value (Titos et al., 2014).

Figure 2 illustrates that the Cazorla et al. (2013) aerosol classification scheme agrees with the expected dominant

aerosol type at continental polluted (EC/OC aerosol), marine polluted (EC/OC aerosol mixed with sea salt), and continental dust/biomass sites (dust and/or EC/OC). On the other hand, the classification scheme assigns dominant aerosol type at remote marine sites and Arctic sites that differ from what would be expected at these sites, given their location and proximity to aerosol sources. Marine clean sites in this analysis (CPR, CPT, GRW, PYE, THD) have a wide spread of AAE values, and although they are all situated on the left side of the plots in Fig. 2, due to a common

low SAE value among the sites, they are not clustered along the AAE axis. All stations in the plot with median SAE values less than or equal to 1.1 are classified as either continental dust/biomass or marine clean, but those classifications cannot be distinguished in the Cazorla et al. (2013) matrix or the modified Cappa et al. (2016) matrix. An improved matrix may include dust, marine aerosol, large coated particles and/or highly processed (aged) particles as possible aerosol types for SAE values less than 1.1. Figure 2(a) shows that marine clean sites exhibit much higher SSA values

than the continental dust/biomass sites with similarly low SAE values, which suggests that the addition of more optical parameters, including SSA, into the clustering analysis could yield more optimized aerosol classification results. Consequently, in the next section, results from a multivariate cluster analysis are used to help reduce ambiguity in aerosol classification and further hone potential aerosol type identification.





## 5.2 Multivariate cluster analysis

In order to infer a more accurate representation of aerosol type using intensive optical properties as an indication of aerosol size/composition and extensive optical properties as an indication of loading, a multivariate clustering analysis is performed. A cluster analysis is the process of statistical grouping that yields 'clusters' with similar characteristics. A

few other studies also implement multi-dimensional clustering as a means of solidifying aerosol property thresholds for different aerosol types (Russell et al., 2010; Omar et al., 2005; Levy et al., 2007). In this study, a cluster analysis is used to determine groups of stations with similar aerosol type based on aerosol optical properties. The clusters are then plotted in a 3D parameter space (AAE vs. SAE vs. $\log(\sigma_{sp})$) as a means of visualizing any spatial patterns that emerge.

The k-means clustering algorithm was run using medians of four aerosol optical property parameters – SAE, AAE,

SSA, and the log of the scattering coefficient ($\log(\sigma_{sp})$) – from hourly averaged records at each monitoring station. The scattering coefficient, $\sigma_{sp}$, is an indication of aerosol loading and is implemented here as an additional parameter to improve the inference of aerosol types. The log of $\sigma_{sp}$ (in Mm$^{-1}$) is used rather than the raw $\sigma_{sp}$ median in order to make the scattering coefficient values more comparable with the magnitude of the optical property values, so the clustering is not dominated by one parameter. While the magnitude of loading ($\sigma_{sp}$) alone does not correspond to a specific aerosol

type (for example, high loadings can be observed for dust, pollution, or biomass burning events) it may act as a secondary indicator of aerosol conditions (i.e., frequency of aerosol type occurrence, loading) and source contributions, so it is included in the clustering analysis.

To run the clustering algorithm, a number of clusters 'k' is selected. Choosing the 'k' initial seed points is inherently subjective – in this analysis, k needs to be small enough such that the number of stations that fall into each cluster

makes for a meaningful grouping, and large enough such that a distinction between station groups is apparent. The algorithm then takes 'k' initial seed points at random and iteratively assigns each point to the nearest cluster centroid taking into account the clustering properties. The next iteration chooses 'k' new seed points and repeats the process until the algorithm converges. In this study, six clusters are selected, creating six unique groups each with similar SAE, AAE, SSA, and $\log(\sigma_{sp})$ characteristics. Each monitoring station was assigned to one of the six clusters produced from the

algorithm, and the groupings were used to further analyze aerosol type and conditions.

Figure 3 shows median optical property values, plotted in a 3D AAE vs. SAE vs. $\log(\sigma_{sp})$ parameter space. Station points are color coded by cluster number, and sized by SSA median values. Not only does the 3D parameter space provide a robust visualization of the clustering results, it provides further insight into an aerosol population than the AAE vs. SAE parameter space used previously, since information on loading and SSA are also visible.

Table 4 shows median AAE, SAE, SSA and $\log(\sigma_{sp})$ values along with interquartile values for each cluster, plus aerosol type and condition (where applicable) based on cluster optical property medians, thresholds from previous literature, and previous knowledge of station characteristics at the sites within each cluster.

In the 3D plot seen in Figure 3, stations that fall within the same cluster number also are located near each other in the three-dimensional parameter space, making for an effective visualization of the relationship between aerosol population

and optical properties. Furthermore, stations in each cluster generally share similar site characteristics and expected aerosol type. Cluster 1 includes MLO, SUM, BRW, ALT and SPL and is characterized by stations with low aerosol loadings (small $\sigma_{sp}$), medium to small aerosol particles (1< SAE<2), and AAE values with a range of 0.5 < AAE < 1.5. All of the stations in Cluster 1 are either Arctic stations (BRW, ALT, and SUM) or remote mountaintop stations (MLO





and SPL). The median optical properties of the cluster correspond with an aerosol type of 'large coated particles' or 'EC/OC' according to Cazorla et al. (2013). On the one hand, the inferred aerosol of 'large coated particles' type agrees well with what aerosols would be expected at remote locations such as these, since most aerosols to reach these sites are well aged and could be coated due to the aging time (Jacobson, 2001). However, this aerosol type may need to be

generalized to 'processed aerosol', to include a larger diversity of aged and processed aerosol that may be sampled at these remote site locations. An aerosol type of 'EC/OC' would also not be surprising at these sites, given the potential for local aerosol source contributions from biomass or fossil fuel burning, though the generally low scattering coefficients at these sites indicate the presence of anthropogenic aerosols is infrequent.

Cluster 2 includes AMY and GSN, the two coastal stations located in South Korea, and is characterized by high aerosol

loadings (high $\sigma_{sp}$), small aerosol particles (SAE~1.5) and carbonaceous aerosols (AAE~1.2). Existing aerosol classification schemes would designate the median optical property values of this cluster as fossil fuel burning aerosols (EC) and/or biomass burning aerosols (OC) (Cazorla et al., 2013; Lee et al., 2012; Russell et al, 2010; Yang et al., 2009). Given the location of these sites within a highly polluted region and occasional local biomass burning, the inferred aerosol types make sense. The relatively low SAE compared to the sites in Cluster 3 can be attributed to the

contribution of sea salt aerosol.

Cluster 3 is comprised of ARN, SGP, FKB, BEO, LLN, PVC, APP, BND and KPS- all of which are continental polluted stations, with the exception of PVC and ARN, which are marine polluted sites characterized by both oceanic air masses and continental polluted air masses. The stations in Cluster 3 have the highest SAE values of all the sites (SAE~2) and thus the smallest particles, and meet the optical property thresholds for fossil fuel and biomass burning aerosols

observed in previous studies (Cappa et al., 2016; Cazorla et al., 2013; Clarke et al., 2007; Lee et al., 2012). What separates Cluster 2 from Cluster 3 is the slightly higher SSA value for sites in Cluster 2, which may indicate a stronger sea salt aerosol signal at the Cluster 2 marine polluted sites that is not present at most sites in Cluster 3 (except ARN). Additionally, aerosol loading at sites in Cluster 2 is substantially higher (factor of 2-3) than at the sites in Cluster 3.

Cluster 4 contains NIM and PGH, both characterized by very high aerosol loadings (high $\sigma_{sp}$), low SAE (SAE~0.5),

indicating large particle, and AAE~1.3. The median optical signature of the aerosol at these monitoring stations is representative of dust aerosol (Costabile et al., 2013; Cazorla et al., 2013; Lee et al., 2012; Yang et al., 2009; Russell et al., 2010; Bahadur et al., 2012) potentially mixed with a black and/or brown carbon (Cappa et al., 2016). Previous studies report that high loadings of dust aerosol are found at both PGH (Kotamarthi, 2013) and NIM (Osborne et al., 2008), validating the inferred aerosol type of this cluster. NIM also experiences some biomass burning aerosol (Osborne

et al., 2008; MacFarlane et al., 2009). PGH experiences fresh biofuel burning daily as well as seasonal pollen and large bioaerosol from the surrounding forest.

Cluster 5 is comprised of just one monitoring site, CPR. Although it is typically undesirable to have a cluster with only one member, the algorithm placed CPR in its own cluster for many choices of 'k' clusters, indicating that indeed it is unique enough to be in its own cluster. Cluster 5 doesn't have interquartile spread on the values since it only has one

station. Cluster 5 (CPR) is characterized by the highest AAE value (AAE median = 2.00), the smallest SAE value (SAE median = 0.28), and a high particle loading. Both marine aerosols and African dust event aerosols have been measured at Puerto Rican sites (e.g., Denjean et al., 2016; Propsero et al., 2014; Kalashnikova and Kahn, 2008; Reid et al., 2003), as well as occasional anthropogenic aerosol (Allan et al., 2008) biomass burning aerosol and volcanic aerosol. Based on median optical properties, the CPR aerosol falls into the dust regime suggested by previous classification schemes



(Costabile et al., 2013; Cazorla et al., 2013; Lee et al., 2012; Yang et al., 2009; Russell et al., 2010; Bahadur et al., 2012), though knowledge of the CPR site suggests the station would also measure sea salt and occasional biomass burning aerosol.

Monitoring stations CPT, GRW, PYE, THD and WLG make up Cluster 6. The cluster is defined by high SSA values (SSA~0.95), low $\sigma_{sp}$ and moderate SAE (SAE~0.96) and AAE (AAE~1.12) values. It is worth noting that the AAE values at stations in this cluster have a large spread, and GRW only fits the criteria of this cluster with its high SSA, but is otherwise anomalous in its optical properties. The optical property values fall within the bounds of multiple aerosol type thresholds suggested by various studies, and thus the aerosol type will be considered mixed. The majority of stations within this cluster (CPT, GRW, PYE, THD) are remote marine sites that may receive occasional dust, biomass burning and/or pollution events amid typical sea salt particle measurements, so a mixed dominant aerosol type might be expected. A slight outlier to this group is WLG, which is a remote mountaintop site. Although sea salt particles are very unlikely to be a constituent of the WLG aerosol, WLG does experience strong dust and pollution events, depending on the season and wind direction (Kivekäs et al., 2009). The main difference between Clusters 5 and 6 is the higher AAE and lower SAE values in Cluster 5 (CPR), indicating a stronger presence of large dust particles and smaller influence of EC/OC pollution at CPR compared to the locations in Cluster 6.

The clusters presented here generally group together sites that are expected to have similar aerosols, and the expected aerosol characterizations generally agree with the aerosol type inferred with the aerosol classification schemes. The method does particularly well with identifying aerosol type at stations with a more or less stable, homogeneous aerosol population, including continental stations sampling EC/OC aerosol (i.e., Clusters 2 and 3), as well as the continental stations sampling high loads of dust aerosol (i.e., Cluster 4). The method also does a fair job at identifying remote Arctic or mountaintop sites (i.e., Cluster 1) that sample large processed particles (due to aging during transport) and occasional instances of local pollution. These methods do not do as well at identifying dominant aerosol type at stations with more complex location and topography, where variable aerosol populations that depend on wind direction and/or occasional extreme aerosol events are not well characterized by median optical properties within the parameter space.

An advantage to the incorporation of $\log(\sigma_{sp})$ into the clustering algorithm and the 3D parameter space plot is that it allows for a more complete picture of aerosol type and conditions at the station. For example, even though the Cazorla et al. (2013) aerosol typing scheme assigns a mixed EC/OC aerosol to both Clusters 1 (remote Arctic and mountaintop stations: ALT, BRW, SUM, MLO, SPL) and 2 (heavily polluted urban coastal sites: AMY, GSN), Fig. 3 shows that these clusters are clearly different, given that Cluster 1 exhibits much lower aerosol loading than Cluster 2. The stations in these clusters are indistinguishable within just the AAE vs. SAE 2D parameter space. Using $\sigma_{sp}$ in the analysis gives further insight into the frequency of occurrence and loading of the inferred aerosol (stations in Cluster 1 measure less EC/OC aerosol than stations in Cluster 2).

There are a few weaknesses to the approaches used thus far in typing aerosols using median optical properties and clustering to reduce ambiguity in the aerosol classification. First, knowledge of station location alone cannot accurately determine the type of aerosols found there (Omar et al., 2005). For example, long-range transport or extreme events may result in aerosols being sampled that are not generally representative of the local geographic region. Second, using a climatological mean or median value of an optical property like SAE or AAE can be misleading in the case that two or more differing aerosols are present at different times over the measurement period. For example, a median SAE value of 1 for a site that measures sea salt (low SAE near 0) over half the measurement period and pollution aerosol




(high SAE near 2) over the other half of the measurement period, does not provide any real information about the aerosol population, since neither aerosol type has an SAE value of 1. In order to address these concerns, an additional analysis using air mass back trajectories is performed as a means of exploring the spread in optical property data at each site. This analysis also allows for multiple aerosol types to be present at any one location.

**5.3 Back trajectory analysis**

The preceding results are derived from application of aerosol typing schemes to median optical properties at multiple stations, a method that depends on the assumption that each site has only a single dominant aerosol type. Many of the sites in this analysis, however, are likely to have a heterogeneous aerosol population with various aerosol types. Backward air mass trajectories are incorporated into the analysis here as a means of both (1) allowing for the consideration of multiple dominant aerosol types at one station and (2) allowing for attribution of likely aerosol source, which can help confirm the practicality of using optical properties to infer aerosol type.

The NOAA Air Resources Laboratory Hybrid Single Particle Lagrangian Integrated Trajectory (HYSPLIT) model (Draxler and Rolph, 2012) was utilized to produce 3-day air mass back trajectories at 6-hour intervals for the entirety of the measurement period at each station. A cluster analysis was performed in HYSPLIT on the back trajectories from individual stations in order to group air masses of similar speed, direction, and altitude. A thorough description of the HYSPLIT cluster analysis methodology can be found in Kelly et al. (2013). The number of back trajectory clusters differs by station, since the selection of cluster numbers is dependent on the individual data set and is somewhat subjective. For this study, and in adherence with typical clustering methodology, a plot of total spatial variance versus number of clusters was used to determine cluster number; the cluster number point just before the total spatial variances increases dramatically is the number of clusters used for analysis at that site. From the cluster analysis, each 6-hour (0, 6, 12, 18 UTC) trajectory was assigned a cluster number and paired with 6-hour averaged aerosol optical property data from the monitoring station for which the back trajectories were produced. For example, the back trajectory at 6 UTC was paired with aerosol optical property data averaged over hours 3-9 UTC. The paired optical property data were then plotted in the AAE vs. SAE plot space and color coded based on back trajectory cluster number, individually for each site. The method described assumes clustered back trajectories may carry similar aerosol type(s) that may be unique from aerosol found in another back trajectory clusters, allowing for variation in aerosols over time at a site that are dependent on the geography from which the air masses arrived at the station.

**5.3.1 Case studies**

Due to a need for brevity, the back trajectory analyses for all 24 stations cannot be presented, so we selected four monitoring stations to present here: Mt. Waliguan, China (WLG), Cape Cod, Massachusetts, USA (PVC), Niamey, Niger (NIM) and Heselbach, Germany (FKB). The 4 sites presented here were chosen to represent cases both where back trajectories helped identify aerosol types and where back trajectories did not elucidate information beyond the initial aerosol classification analysis using median optical properties. Shown for each of the 4 stations (Figs. 4-7) are a map of mean back trajectory paths for each cluster, a plot of trajectory height vs. backward time (color coded by trajectory cluster number), and a plot of AAE vs. SAE properties for 6-hour averaged optical property data, color coded by paired trajectory cluster number, and overlaid by median optical property values of each cluster in the largest color-coded point. If a station's dominant aerosol type differs with air mass origin, these plots can elucidate a station's various aerosol types.



### 5.3.1.1 Mt. Waliguan, China

The back trajectories at Mt. Waliguan (WLG) were grouped into 4 clusters in HYSPLIT, as shown in Figure 4. Cluster 1 contains ~33% of the site's back trajectories and has origins to the west of the station near Northern Pakistan and traveling through Western China; Cluster 2 contains ~30% of the site's back trajectories and has origins (on average) to

5 the west of the station in rural China; Cluster 3 in contains ~33% of the site's back trajectories and has origins very near the site itself and slightly to the east; and Cluster 4 contains ~3% of the site's back trajectories and has origins to the far northwest of the station, traveling to the station at high altitudes from rural Russia. AAE values are similar for each trajectory cluster, though SAE values vary. Furthermore, the median aerosol optical property values from each of the trajectory clusters are unique, suggesting a variety of aerosol types using thresholds from previous literature (Cazorla et

10 al., 2013; Costabile et al, 2013). The optical properties from the aerosols in back trajectory Cluster 1 (from deserts in Northern Pakistan and Western China) imply a dust mixture. Lower SAE values mean the aerosols from this trajectory cluster are larger, and AAE values near and above 1.5 likely mean dust and/or carbonaceous aerosol mixture (Cazorla et al., 2013). These results support those of Che et al. (2011) and Kivekäs et al. (2009), which cite deserts as aerosol sources from western wind sectors at WLG. Clusters 1 and 2 are most similar in terms of median SAE and AAE values,

15 though the map shows that Cluster 1 trajectories traveled farther in the 3-day period, and thus had faster wind speeds. Cluster 2 and cluster 3 have mean trajectory paths that are relatively short, and thus associated with low wind speeds. This means that these clusters are likely to be more influenced by local aerosol sources. The optical properties of the aerosols from back trajectory Cluster 3 coming from the east suggest black carbon given the AAE value near 1. This is in agreement with findings of Kivekäs et al. (2009) that show increased particle concentrations from the east of the

20 WLG station indicated anthropogenic pollution. Cluster 4 looks quite different than the other trajectories, and has median optical properties indicative of dust (Cazorla et al., 2013), which makes sense given the trajectory cluster's origin to the northwest of the site (Che et al., 2011; Kivekäs et al., 2009).

### 5.3.1.2 Niamey, Niger

The back trajectories at Niamey (NIM) were grouped into 3 clusters in HYSPLIT, as shown in Figure 5. Cluster 1

25 contains slightly over half (~53%) of the back trajectories, with air-mass trajectories reaching the site (on average) from the south/southwest, and traveling at a relatively low altitude over populated regions. Cluster 1 differs from Clusters 2 and 3 in that it has a lower median AAE value and higher median SAE value. Given the optical properties of the trajectory cluster 1, along with the knowledge of anthropogenic activities in the source region, the likely dominant aerosol during those trajectories is a biomass burning/soot aerosol mixture (e.g., Osborne et al., 2008; MacFarlane et al.,

30 2009). Clusters 2 and 3 constitute slightly less than half (~46%) of the back trajectories at NIM and originate (on average) from the north and northeast of the site. In Fig. 5, the median optical property values of Clusters 2 and 3 are nearly indistinguishable. For these two clusters, the small SAE values and AAE values above ~1.5 suggest dust (Cazorla et al., 2013; Lee et al., 2012; Yang et al., 2009). Previous observations by Osborne et al. (2008) noted dust during northerly flow due to the proximity of the Sahara desert to the north/northeast of the site, as did MacFarlane et

35 al. (2009). NIM provides a good example of trajectory analysis elucidating two dominant aerosol types that were obscured when only the climatological medians of AAE and SAE values were evaluated. However, it should be noted that local sources and meteorological conditions also have a large influence on aerosol at the site, in addition to trajectory sources.





### 5.3.1.3 Cape Code, Massachusetts, USA

Back trajectories at Cape Cod (PVC) were clustered into 3 groups in HYSPLIT, as shown in Figure 6. Cluster 1 contains almost half (~49%) of the trajectories and originates (on average) to the south and southeast of the Cape Cod site along the heavily populated eastern U.S. seaboard. Cluster 2 contains ~43% of the trajectories and (on average) travels to the monitoring station from the Northwest over eastern Canada. Cluster 3 contains only ~8% of trajectories and comes to the station from over the North Atlantic. Cluster 3 is distinct from Clusters 1 and 2 with the lowest SAE value (largest particles), and given its source region suggests at least partial marine sea salt aerosols. Clusters 1 and 2, on the other hand, with continental source regions and optical properties indicative of elemental and organic carbon suggest anthropogenic aerosols. Due to its proximity to both the ocean and large cities like Boston, it is unsurprising that the site measures both marine and urban aerosols, depending on the wind direction. The pairing of back trajectory analysis with optical property classification gives a more detailed picture of the multiple aerosol populations at PVC, in accord with other aerosol research done at the site (Titos et al., 2014). Since the back trajectories from over the Atlantic make up such a small portion of the air masses that arrive at PVC, this could explain why this station clusters with continental polluted stations instead of marine polluted stations in the first cluster analysis of this study in Sect. 5.1.

### 5.3.1.4 Heselbach, Black Forest, Germany

Back trajectories at FKB group into 2 clusters (Figure 7), each containing approximately half of the back trajectories. Back trajectories associated with Cluster 1 typically originated from the northwest over the North Atlantic and are associated with higher wind speeds and longer distance transport than those in cluster 2. Cluster 2 tended to travel shorter distances in reaching the site, with mean back trajectories originating from the east of the station in Southern Germany, as shown in Figure 7. Despite the very different geographical origins of the two air mass clusters, and very different wind speeds (on average), both trajectory groups have similar median optical property signatures, and suggest an aerosol type of elemental and organic carbon based on thresholds given in previous literature (Cazorla et al., 2013; Yang et al., 2012; Costabile et al., 2013; Lee et al., 2009). The similarity of aerosol properties between the two trajectory clusters arriving at FKB suggests that FKB measures aerosols that are regionally representative aerosol of Western Europe. Previous analysis of FKB data shows that the site is dominated by anthropogenic aerosol (Jefferson, 2010). Due to the homogeneity of the aerosol population at the FKB site, back trajectory analysis does not provide any additional information useful for aerosol typing.

### 5.3.2 All stations

A broader understanding of the link between back trajectory clusters, aerosol optical property measurements and aerosol type can be gained by collectively analyzing all trajectory clusters at all stations, rather than looking at stations individually. Here each trajectory cluster from every site is classified based on where the trajectory originated and the geography over which the air mass traveled, then trajectory clusters from *all* stations are plotted in one AAE vs. SAE plot space. The classifications include continental Arctic, continental dust, continental dust/polluted, continental polluted, polluted marine, and remote marine. A trajectory cluster is classified as continental Arctic if it passes over land north of 60N latitude, continental dust if it passes over remote desert, continental dust/polluted if it passes over populated desert regions with anthropogenic influence, continental polluted if it passes over populated land, polluted marine if it passes over populated coastal regions with anthropogenic influence, and remote marine if it passes over clean, unpopulated ocean regions. Table S6 in supplemental materials details classifications of each trajectory cluster at all stations. There is unavoidable subjectivity in this classification method, for a few reasons. For one, some trajectories





travel over geography that falls into one or more of the classifications chosen for the analysis. In these cases, other factors, such as underlying geography and typical site aerosol populations, were considered to make the more nuanced classifications. Back trajectory analysis of aerosol type are needed to account for air mass dispersion, aerosol wet and dry deposition, cloud processing, as well as additional sources added at low altitudes and locally. A good example of this is long-range transport of African dust over the Atlantic Ocean. A 3-day back trajectory may not be sufficient to identify long-range dust transport from the African continent. Here, delineation of dust from marine aerosol is ambiguous. More information on the aerosol composition and hygroscopicity is needed for more conclusive aerosol identification. The authors acknowledge this weakness of the methodology and its inherent uncertainty and subjectivity.

Median values of optical properties from each trajectory cluster at all sites are presented in Figure 8. There are some clear spatial patterns that emerge when visualizing the trajectory cluster classifications and the median optical properties in the AAE vs. SAE plot space. The majority of continental polluted trajectory clusters group tightly in the area of the plot that would be classified as EC/OC aerosol by the Cazorla et al. (2013) matrix. This is similar to earlier findings in this paper where continental polluted sites were aggregated at higher SAE (smaller size) and at AAE values in the range of ~1-1.5. Trajectories classified as polluted marine show a similar range of AAE values as the continental polluted trajectories, though with lower SAE values, indicative of large sea salt mixed with organic carbon. Trajectory clusters classified as continental dust are best defined by AAE values greater than 1.4, though are poorly defined by SAE values due to the large variance in SAE for those clusters. Continental dust/polluted trajectory clusters are more or less tightly defined by AAE values between 0.9-1.4, and SAE values between 0.5-1.2, though it is hard to draw significant conclusions about this trajectory type, since only three trajectories meet this classification. Trajectories identified as continental Arctic are not well defined in this plot space. Both AAE and SAE values of this trajectory type are variable, though median SSA values for this trajectory class are more similar, and are close to 0.95.

The range of Arctic optical properties most likely stems from the seasonal transport of European and Siberian continental aerosol to the sites in the winter and spring, contrasted with sea salt from open water in the summer. Remote marine trajectories are the least well defined of all the trajectory cluster classes, with highly variable optical properties. Remote marine trajectories show AAE values that range anywhere from 0-2.2, with SAE values slightly more defined at a range of -0.4-1.2. Median SSA values are, however, quite similar within more remote marine trajectories, with high values near 0.96 indicating a whiter aerosol such as sea salt.

There are some clear outliers within trajectory classification groups that may be explained by misclassification of trajectories. For example, the points labeled 1 and 2 in Figure 8 are back trajectories from CPR, both with 3-day paths that travel only over the Atlantic Ocean. Although the trajectory classification methodology yielded a class of remote marine for those specific trajectories (the air masses only traveled over unpopulated ocean regions for 3 days before reaching the site), previous studies suggest that these air masses could be heavily influenced by African dust events (Denjean et al., 2016; Kalashnikova and Kahn, 2008; Reid et al., 2003). If indeed the dominant aerosol type in these back trajectories was dust, this would fit in much more neatly to previous dust classification schemes (i.e., Lee et al., 2012; Clarke et al., 2007; Yang et al., 2009) and the Cazorla et al. (2013) matrix.

By classifying back trajectory clusters from all station locations, and including them in the optical property plot space, we get a clearer idea of what types of trajectories, and thus likely aerosol type, are well defined by median optical properties, and those that are poorly defined by median optical properties. Continental polluted and marine polluted trajectories have median optical parameters that are well defined, and visually cluster in the plot space. Continental dust



and continental dust/biomass are somewhat well defined by optical properties in the plot space. Continental Arctic trajectories appear to be well defined by AAE, with all cluster AAE values around 1, though the trajectories are not well defined by SAE, which shows a larger range. The remote marine trajectory cluster (presumably clean air masses) is poorly defined by optical properties and thus is not easily visualized in the plot space.

To our knowledge, few previous studies have classified remote marine aerosol (only Costabile et al. (2013) classified a coarse marine mode in the suburbs of Rome, Italy), and no previous studies have classified continental Arctic aerosols using an aerosol classification matrix. Our findings show that at these site types, typing schemes that use aerosol optical properties need more detailed analysis that account for seasonal variability and local sources. Using aerosol optical parameters to infer aerosol type works well for certain types of aerosol that fit neatly into matrices like that from

Cazorla et al. (2013), including EC/OC aerosol and dust. Marine aerosol, processed aerosol, and mixed aerosol populations are much more poorly defined by optical properties, and do not fit cleanly in existing matrices without overlap with different aerosol types.

**6. Discussion**

Application of previous aerosol classification schemes to the aerosol optical property data from stations in the NOAA

Federated Aerosol Network generally yields a dominant aerosol type that would be expected at that site location. The classification schemes do particularly well at inferring aerosol type from optical properties at continental sites that measure carbonaceous aerosols, but do not do as well at sites with more complex topography (e.g., mountaintop, coastal) that measure a more heterogeneous aerosol population that changes with wind direction. Including median optical parameters from multiple stations on one AAE vs. SAE plot allows for comparison of dominant aerosol type at

many sites, though the use of median optical properties makes the most sense for sites with a homogenous aerosol population. The single AAE vs. SAE plot can provide ambiguous results for sites with a heterogeneous aerosol population.

The two aerosol classification methods (Sect. 5.1 and 5.2) had varying degrees of success. The first method, a multivariate cluster analysis, generated groups of monitoring sites with similar AAE, SAE, SSA and $\log(\sigma_{sp})$ values.

The first classification scheme was applied to median optical properties from all station data within each cluster to produce a new aerosol type for stations within that cluster. One advantage to this approach is that the inclusion of $\log(\sigma_{sp})$ in the clustering analysis, and subsequent visualization of station clusters in the AAE v. SAE v. $\log(\sigma_{sp})$ 3D parameter space, provides insight not only into a cluster's aerosol type. This approach also provides insight as to how aerosol loading (and thus site conditions) differ between clusters. Although the AAE and SAE aerosol typing schemes

yield similar inferred aerosol type of carbonaceous aerosol for both remote Arctic/mountaintop sites and continental sites, the notable difference in $\log(\sigma_{sp})$ values among these dissimilar stations defines the separate clusters. An anticipated advantage to the multivariate cluster analysis was that it would help to reduce ambiguity in results of aerosol typing schemes, though this was not the case with every cluster. Rather than falling more surely within the optical property thresholds of one aerosol type, the median optical properties of a few clusters still fell on the cusp of two or

more aerosol type thresholds. This left the aerosol type of some clusters uncertain, particularly for clusters with coastal and/or remote sites.

The second method (Sect. 5.3), pairing 6 h averaged optical properties with corresponding back trajectories, provided more detailed insight into the aerosol population at an individual station. This method allowed for typing of multiple aerosols related to different air masses. At stations where aerosol populations are diverse and varying, such as NIM





(dust and biomass burning), WLG (dust, pollution, free troposphere long-range transport aerosol), and PVC (marine aerosol and pollution), the different aerosol types that were previously obscured using the site's median optical properties were more apparent when using the trajectory cluster approach. At stations where aerosol populations are homogeneous (like FKB (regional pollution)), no new information on aerosol type was gained. Consolidating all

trajectory clusters and corresponding classifications into one plot space (Sect. 5.3.2) allowed us to see a large variety of back trajectory and likely aerosol type, and confirmed previous findings from the paper- that some trajectory classes (like continental polluted and marine polluted) are well defined by a unique range and combination of optical properties, while other trajectory classes (like remote marine and continental Arctic), have highly variable ranges and combinations of SAE, AAE and SSA, and are thus less likely to be typed by aerosol classification schemes using only

optical parameters.

The application of varying classification methods gave satisfactory inferences of some aerosol types, in great part due to the quality of previously developed aerosol classification schemes. Despite the differences in optical property thresholds presented from each scheme, many of the schemes' thresholds do have large overlap, making it easy to affirm inferred aerosol type with multiple schemes. Many typing schemes provided satisfactory aerosol typing results

for fossil fuel burning aerosol, biomass burning aerosol and dust (Lee et al., 2012; Yang et al., 2009; Bahadur et al., 2012; Russell et al., 2010), though fewer schemes were available to type large coated particles (Cazorla et al., 2013), sea salt (Costabile et al., 2013) and mixed aerosol (Cazorla et al., 2013). Perhaps the most useful typing scheme was that of Cazorla et al. (2013), which provided thresholds for typing mixed aerosol and large coated particles. The Cazorla et a. (2013) scheme also delineated the entirety of the AAE vs. SAE plot space, leaving no combination of optical

property values without a category.

A major missing piece of the currently available aerosol classification methods is identification and validation of optical property thresholds to identify sea salt aerosol. To the authors' knowledge, only one study includes marine aerosol identification; Costabile et al. (2013) provide values of SSA > 0.95, SAE < 0.5, dSSA=0-0.05 and AAE > 2 for coarse marine mode aerosol. Many studies ignore the contribution of sea salt altogether (or do not use data that would have sea

salt aerosol contribution), while other studies do not include sea salt aerosol in their typing scheme because sea salt has negligible absorption and thus poorly defined AAE (Russell et al., 2010). Since sea salt aerosols are dominated by large particles, there is a general consensus that marine particles are characterized by low SAE values and high SSA values (Russell et al., 2010; Costabile et al., 2013; Smirnov et al., 2002; Dubovnik et al., 2002). Of the 24 stations analyzed in this study, sea salt aerosol is expected at CPT, CPR, GRW, PYE, THD, and to a lesser extent at ARN, AMY, GSN and

PVC. With the exception of ARN, AMY, GSN and PVC, which often measured polluted air masses (see scattering coefficient values for these four stations in Table 3 and back trajectories for PVC), these coastal stations have median values of SAE < 1 and SSA > 0.95. Median values of AAE, however, range from 0.5-2.0. Further back trajectory analysis (not shown here) relating air masses of oceanic origin at these sites to aerosol optical properties does not show specific patterns in AAE values for marine aerosols. Although no new marine aerosol typing information is included

here, the authors do encourage consideration of SAE and SSA thresholds for sea salt to be included in future aerosol classification analyses. Furthermore, the authors acknowledge that although no sea salt aerosol types are designated here explicitly at coastal stations, some of the aerosol types are likely sea salt aerosol mixed (however slightly) with some absorbing component. Cappa et al. (2016) in some ways account for sea salt aerosol by changing the categorization in the lower left of the box to "large particle/lower absorption mix" although they also suggest this

regime could be represented by "large black particles".



Although this study generally affirms existing aerosol typing schemes, the results here are only applicable given certain conditions and for specific aerosol types. One stipulation of this analysis is that results were compared to aerosol typing schemes from studies that used optical property data from in situ surface measurements, aircraft campaigns, and AERONET measurements. There are few studies (e.g., Cappa et al., 2016) that evaluate the differences that may exist in aerosol typing schemes/thresholds based on the type of data (in situ vs. remote sensing, column vs. point, dry vs. ambient measurements) used. The difference in RH between dry (most in situ surface) and ambient (AERONET) measurements could have some effect on the determined thresholds. A higher RH would decrease SAE (larger aerosol), SSA thresholds might shift up (whiter aerosol), scattering coefficients would get larger, and AAE might change due to coating on absorbing particles. Future analysis comparing dry and ambient aerosol typing schemes would be useful for determining the validity of the comparisons made in this study.

An additional caveat in the parameter clustering analysis and back trajectory cluster analysis is the presence of externally mixed aerosol with size-dependent composition that renders the analysis ambiguous for a given aerosol class. Future work on this would add much needed information to the subject of aerosol typing from optical properties.

Another limitation to the classification analyses presented here is that aerosol aging during transport can influence aerosol type. A study by Devi et al. (2016) shows that prior to atmospheric aging, mobile sources and biomass burning sources can have relatively high (~1.2-2.0) AAE values; however, after aging during transport (~1-2 days), the brown carbon signal can go away, reducing the AAE value. There may be a point when source information from aerosol intensive optical properties can be lost during transport. In that case, aerosol classification schemes may no longer be applicable.

There are still many ways in which this analysis can be expanded. The incorporation of aerosol shape into the typing analysis could be helpful, particularly in determining the differences between particles with similar optical properties. Further stratification of the measurement data by season, time of day, composition or hygroscopicity would elucidate more about the variability of aerosol type with time. And finally, more analyses of stations that have concurrent chemistry measurements and aerosol optical property measurements could help verify existing aerosol classification schemes (e.g., Cappa et al. (2016) and Costabile et al. (2017)).

### 7. Conclusion

Surface in situ aerosol optical properties obtained at 24 stations in the NOAA Federated Network were used to classify aerosol type at the site, using aerosol classification schemes from the literature, cluster analyses, and general knowledge of station location and characteristics. The monitoring sites utilized for the analysis offered a diverse range of station locations and aerosol types, providing a look at fossil fuel burning, biomass burning, sea salt, dust as well as regionally mixed aerosols observed at various continental sites. Plotting station optical property medians in an AAE vs. SAE plot space, overlaid by the Cazorla et al. (2013) classification matrix, for the most part yielded inferences of aerosol types that were to be expected based on knowledge of the monitoring station location. A handful of stations, however, yielded unexpected results that appeared uncharacteristic of the site, which indicated a need for a different visualization or analysis method. Furthermore, the interquartile values of the optical properties from each station in an AAE vs. SAE parameter space showed that there is often large variability in optical properties at any given location, suggesting that a single 'dominant aerosol type' is not realistic at all stations.

A multivariate cluster analysis was performed as a means of grouping together monitoring sites with not only similar





aerosol type, but similar site conditions (frequency of aerosol type, loadings, proximity to source, location, etc.). The multivariate cluster analysis yielded 6 clusters of stations with similar median AAE, SAE, SSA and $\log(\sigma_{sp})$ values. Sites that grouped within the same cluster most often had similar expected aerosol types that aligned with the aerosol type predicted by the aerosol typing scheme. Incorporation of the scattering coefficient into the multivariate cluster

analysis improved the inference of aerosol type and conditions (i.e., aerosol loading, source) from optical property measurements.

In order to further explore the complexity of aerosol populations and allow for multiple aerosol types at some sites, an additional analysis was presented using air mass back trajectories. Air mass back trajectories were clustered based on similar direction, altitude and speed, and these clusters were paired with optical property data and plotted in the AAE

vs. SAE parameter space. More detailed results from 4 of the 24 stations – WLG, NIM, PVC and FKB – were discussed in order to show the range of success (or lack thereof) of this approach. At complex sites like WLG, NIM, and PVC, multiple dominant aerosol types emerged, unique to different clusters of air mass back trajectories. The classification of numerous aerosol types, along with the information from the back trajectory clusters on how often those aerosol types were measured, allowed for a more complete picture of the heterogeneous aerosol populations at those sites. In the case

of FKB, only one aerosol type is inferred in each of the different trajectory clusters, suggesting a homogenous aerosol population that is readily predicted by the simpler analysis of just the median optical properties in the AAE vs. SAE parameter space.

Combining back trajectory clusters and classifications from all 24 sites showed that comparing optical characteristics with trajectory characteristics yields results that further inform aerosol typing schemes. While all trajectory clusters that

were classified as marine polluted or continental polluted had optical properties that were well defined, other trajectory clusters classified as continental Arctic or remote marine had highly variable optical parameters that were not informative in aerosol typing.

This study has further assessed existing aerosol typing schemes, provided additional methods that can be implemented to reduce ambiguity in typing schemes, elucidate aerosol conditions that accompany aerosol type, and allow for

identification of multiple aerosol types at one site. Furthermore, this paper highlighted the need for further analyses and suggests specific ideas for future work needed to progress and refine aerosol typing schemes that infer aerosol type from optical properties.

**Data availability**

Data for AMF sites are available from the DOE/ARM website (http://www.arm.gov). Data from all other sites (except

WLG) are available from the World Data Center for Aerosols (http://ebas.nilu.no/). WLG data are available from Junying Sun at CAMS.

**Acknowledgements**

The authors would like to acknowledge the U.S. Department of Energy as part of the Atmospheric Radiation

Measurement (ARM) Climate Research Facility for use of data from Southern Great Plains, Oklahoma, USA, as well as data from the ARM Mobile Facility that was stationed at Heselbach, Germany; Graciosa Island, Azores, Portugal; Niamey, Niger; Nainital, India; Cape Cod, Massachusetts, USA; and Point Reyes, California, USA. We would like to thank Derek Hageman for his extensive technical assistance in obtaining and archiving the data from all stations in this



paper. The aerosol measurements at ALT are operated by Environment and Climate Change Canada, and instrument maintenance/calibrations by Dan Veber, the operators, staff of Canadian Forces Services in maintenance and operation of the ALT site. The monitoring at ARN was partially supported by the Spanish Ministry of Science and Technology (MINECO) through the Project CGL2014-55230-R and the European Union's Horizon 2020 research and innovation

programme under grant agreement N° 654109 (ACTRIS2). The authors would like to acknowledge the China Meteorological Administration for use of data from Mt. Waliguan, China. This work was partly supported by grant from National Basic Research Program of China (2014CB441201), the National Natural Science Foundation of China (41675129). Sang-Woo Kim was supported by the KMA R&D program under Grant KMIPA 2015-2011. The measurements at BEO are supported by the EU Project ACTRIS2 H2020-INFRAIA and the Bulgarian Academy of

Sciences. The measurements at LLN are supported by the Environmental Protection Administration of Taiwan.

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





**Table 1 Aerosol optical property thresholds used to determine aerosol type in previous studies. Values in parentheses represent standard deviations, when provided**

| Study | Measurement Type | Dust | Fossil Fuel Burning | Sea salt | Biomass Burning |
|---|---|---|---|---|---|
| Bahadur et al. (2012) | AERONET | $AAE_{440/675nm} \sim 2.2\ (\pm0.50)$ <br> $SAE_{440/675nm} < 0.5$ | $AAE_{440/675nm} \sim 0.55\ (\pm 0.24)$ <br> $SAE_{440/675nm} > 1.2$ <br> *(referred to as BC/EC/soot)* | | $AAE_{440/675nm} \sim 4.55\ (\pm 2.01)$ <br> $SAE_{440/675nm} > 1.2$ <br> *(referred to as OC)* |
| Cazorla et al. (2013) | AERONET and aircraft campaign | $AAE_{440/675nm} > 1.5$ <br> $SAE_{440/675nm} < 1$ | $AAE_{440/675nm} \leq 1$ <br> $SAE_{440/675nm} > 1.5$ <br> *(referred to as EC dominated)* | | $AAE_{440/675nm} \geq 1.5$ <br> $SAE_{440/675nm} > 1.5$ <br> *(referred to as OC dominated)* |
| Russell et al. (2010) | AERONET and aircraft campaign | $AAE = 1.5\text{-}2.5$ <br> $EAE = 0.2\text{-}1$ | $AAE = 0.8\text{-}1.5$ <br> $EAE = 1.5\text{-}1.8$ | | $AAE = 1\text{-}1.7$ <br> $EAE = 1.8\text{-}2$ |
| Clarke et al. (2007) | Aircraft campaign | | $AAE_{470/660nm} \sim 1.1$ <br> *(referred to as pollution)* | | $AAE_{470/660nm} \sim 2.1$ |
| Costabile et al. (2013) | Surface in situ | $AAE_{467/660nm} \sim 2$ <br> $SAE_{467/660nm} < 0.5$ <br> $SSA_{530nm} > 0.85$ <br> *(referred to as coarse dust mode, CDM)* | $AAE_{467/660nm} < 1.5$ <br> $SAE_{467/660nm} \sim 4$ <br> $SSA_{530nm} < 0.8$ <br> *(referred to as soot mode, STM)* | $AAE_{467/660nm} > 2$ <br> $SAE_{467/660nm} < 0.5$ <br> $SSA_{530nm} > 0.95$ <br> *(referred to as coarse marine mode, CMM)* | $AAE_{467/660nm} < 2$ <br> $SAE_{467/660nmmm} \sim 1\text{-}3$ <br> $SSA_{530nm} < 0.85$ <br> *(referred to as biomass burning smoke mode, BBM)* |
| Lee et al. (2012) | Surface in situ | $AAE_{450/700nm} \sim 1.2\text{-}1.7$ <br> $SAE_{450/700nm} \sim 0\text{-}1.2$ <br> *(referred to as PD)* | $AAE_{450/700nm} \sim 1\text{-}1.5$ <br> $SAE_{450/700nm} \sim 1.4\text{-}1.8$ <br> *(referred to as P2)* | | $AAE_{450/700nm} \sim 0.8\text{-}1.4$ <br> $SAE_{450/700nm} \sim 0.8\text{-}1.5$ <br> *(referred to as P1, higher in OC than P2)* |
| Yang et al. (2009) | Surface in situ | $AAE_{370/950nm} \sim 1.82\ (\pm0.90)$ <br> $SAE_{450/700nm} \sim 0.59\ (\pm 0.41)$ <br> $SSA_{550nm} \sim 0.9\ 0.8\ (\pm 0.04)$ | $AAE_{370/950nm} \sim 1.46\ (\pm 0.15)$ <br> $SAE_{450/700nm} \sim 1.39\ (\pm 0.20)$ <br> $SSA_{550nm} \sim 0.8\ (\pm 0.05)$ <br> *(referred to as coal pollution)* | | $AAE_{370/950nm} \sim 1.49\ (\pm 0.08)$ <br> $SAE_{450/700nm} \sim 1.52\ (\pm 0.18)$ <br> $SSA_{550nm} \sim 0.89\ (\pm 0.01)$ |





**Table 2 Monitoring Site Locations and Descriptions. Rows in bold indicate stations that are part of the ARM Mobile Facility (AMF) program and are temporary measurement sites.**

| Station Abbreviation | Station Location | Latitude Longitude Altitude (m asl) | Absorption Instrument* | Measurement Dates | Site Classification | Site Description (and references) |
|---|---|---|---|---|---|---|
| ALT | Alert, Canada | +82.45 -62.52 210 | PSAP-3W | 2012-2013 | Arctic | Remote Arctic site, situated away from major anthropogenic and industrial areas, and the most northerly site in the network. (Sharma et al., 2002) |
| AMY | Anmyeon-do, South Korea | +36.54 +126.33 45 | CLAP-3W | 2012-2013 | Polluted Marine | Polluted marine site that receives both continental and marine air masses, located on Anmyeon Island off the coast of South Korea (Park et al., 2010) |
| APP | Boone, North Carolina, USA | +36.2 -81.7 1100 | PSAP-3W | 2012-2013 | Continental Polluted | Semi-rural continental site, located in the Appalachian Mountains, a region high in biogenically-derived aerosol (Sherman et al., 2015) |
| ARN | El Arenosillo, Spain | +37.10 -6.73 41 | CLAP-3W | 2012-MAY-15 to 2013 | Marine Polluted | Located near the Atlantic Ocean and Huelva City. Site is located in protected coastal area of Doñana National Park and experiences episodes of desert dust and pollution (Toledano et al., 2007) |
| BEO | BEO-Moussala, Bulgaria | +42.18 +23.59 2925 | CLAP-3W | 2012-JUN-03 to 2013 | Continental Polluted, Mountaintop | The Basic Environmental Observatory (BEO) sits atop Moussala Peak, the tallest point on the Balkan Peninsula. Given the site's altitude, it is considered to be in the free troposphere and more or less unperturbed by regional pollution sources (Angelov et al., 2011) |
| BND | Bondville, Illinois, USA | +40.05 -88.37 230 | CLAP-3W | 2012-2013 | Continental Polluted | Anthropogenically influenced rural site located in Champaign County, Illinois, USA, near soy and corn farms south of Bondville (Delene and Ogren, 2002; Sherman et al., 2015) |
| BRW | Barrow, Alaska, USA | +71.32 -156.6 11 | CLAP-3W | 2012-2013 | Arctic | Coastal Arctic site 3 km from Arctic Ocean, located north of the Arctic Circle near the small town of Barrow. Though the site is remote, drilling activities nearby may influence aerosol populations (Bodhaine, 1995) |
| CPR | Cape San Juan, Puerto Rico | +18.48 -66.13 17 | CLAP-3W | 2012-MAR-30 to 2013 | Marine Polluted | Marine site, located on the northeast edge of the Caribbean island of Puerto Rico on Las Cabezas de San Juan nature reserve. Prone to African desert dust episodes (Allan et al., 2008) |
| CPT | Cape Point, South Africa | -34.35 +18.49 230 | PSAP-3W | 2010-2011** | Marine Clean | Marine site, located on the southwest tip of South Africa. Site is influenced by remote marine air and polluted and/or dusty continental air (Brunke et al., 2004) |
| **FKB** | **Heselbach, Germany** | +48.54 +8.40 511 | PSAP-3W | 2007-MAR-23 to 2007-DEC-31 | Continental Polluted | Continental site in the Black Forest region of Germany surrounded by coniferous trees. The site is in the agricultural Murg valley, and experiences heavy precipitation and influence from anthropogenic industrial activities (Jefferson, 2010) |
| **GRW** | **Graciosa Island, Azores, Portugal** | +39.09 -28.03 | PSAP-3W | 2009-APR-18 to | Marine Clean | Marine site located on the remote Azores Islands surrounded by the Atlantic Ocean. Site may be influenced at times by local pollution and African desert dust episodes |


| | | 15.24 | | 2010-DEC-31 | | (Jefferson, 2010) |
|---|---|---|---|---|---|---|
| GSN | Gosan, Jeju Island, South Korea | +33.28 +126.17 72 | CLAP-3W | 2012-2013 | Marine Polluted | Coastal site located on the western edge of Jeju Island, and prone to influence from marine aerosols, anthropogenic pollution, and long-range Asian desert dust (Kim et al., 2005) |
| KPS | K-puszta, Hungary | +46.96 +19.58 125 | CLAP-3W | 2012-2013 | Continental Polluted | Continental site located in the Hungarian Great Plain 70 km southeast of Budapest. Measures regional background air, and although it is situated as remotely as possible, is still influenced by biomass burning aerosol from home heating in the winter (Ion et al., 2005) |
| LLN | Lulin, Taiwan | +23.47 +120.87 2862 | CLAP-3W | 2012-2013 | Continental Polluted, Mountaintop | High altitude site influenced by air masses from polluted biomass and industrial continental Asian sources, as well as clean marine regions (Wai et al., 2008) |
| MLO | Mauna Loa, Hawaii, USA | +19.54 -155.58 3397 | CLAP-3W PSAP-3W | 2012-2013 | Marine Polluted, Mountaintop | High altitude site on the northern side of the Mauna Loa volcano on the big island of Hawaii. Distinct diurnal patterns in upslope/downslope air flow, with minimal influence from regional aerosol sources (Bodhaine, 1995) |
| **NIM** | **Niamey, Niger** | +13.48 +2.18 205 | PSAP-3W | 2005-DEC-01 to 2006-DEC-31 | Continental Dust/Biomass | Continental site susceptible to biomass burning and African desert dust, prone to high heat and heavy rains in the monsoon season (Liu and Li, 2014) |
| **PGH** | **Nainital, India** | +29.36 +79.46 1951 | CLAP-3W | 2011-JUN-09 to 2012-MAR-27 | Continental Dust/Biomass | Continental site located in the Ganges Valley in the remote foothills of the Himalayas. Biomass burning, dust, and growth in nearby industrial activities, sporadically influence the site (Liu and Li, 2014) |
| **PVC** | **Cape Cod, Massachusetts, USA** | +42.07 -70.20 1 | CLAP-3W | 2012-JUL-16 to 2013-JUN-24 | Marine Polluted | Marine site on a peninsula of Massachusetts reaching into the Atlantic Ocean. Site is also near major urban areas, including Boston, Massachusetts and Providence, Rhode Island, and is thus influenced by both polluted and clean air masses (Titos et al., 2014) |
| **PYE** | **Pt. Reyes, California, USA** | +38.09 -122.96 5 | PSAP-3W | 2005-MAR-21 to 2005-SEP-15 | Marine Clean | Marine site on the California coast north of San Francisco. Air masses from the west are strictly maritime, while air masses from the north, south, and east are influenced by continental pollution (Berkowitz et al., 2005) |
| SGP | Southern Great Plains, Oklahoma, USA | +36.61 -97.49 315 | CLAP-3W | 2012-2013 | Continental Polluted | Rural continental site located near wheat fields and cattle pastures southeast of Lamont, Oklahoma. There are no large urban areas nearby, but point sources, like power plants and oil operations, influence the site occasionally (Delene and Ogren, 2002; Sherman et al., 2015) |
| SPL | Storm Peak, Colorado, USA | +40.45 -106.73 3220 | CLAP-3W | 2012-2013 | Continental Polluted, Mountaintop | High altitude forested site in the Rocky Mountains of northwestern Colorado. Located near the town of Steamboat Springs and agricultural Yampa Valley, though the station frequently measures uncontaminated free troposphere. (Borys & Wetzel, 1997) |
| SUM | Summit, Greenland | +72.58 -38.48 | CLAP-3W | 2012-2013 | Arctic, Mountaintop | Arctic station atop the Greenland Ice Sheet. Remote and clean, with occasional influence from long-range biomass and |



| | | | | | | |
|---|---|---|---|---|---|---|
| | | 3238 | | | | industrial pollution (Hagler et al., 2007) |
| THD | Trinidad Head, California, USA | +41.05 -124.15 107 | CLAP-3W | 2012-2013 | Marine Clean | Marine site on the northern California coast, with Pacific Ocean to the west and redwood forests to the east. Though maritime airflow is predominant, some anthropogenic influences from other airflows is observed (Oltmans et al., 2008) |
| WLG | Mt. Waliguan, China | +36.28 +100.90 3816 | PSAP-3W | 2012-2013 | Continental Dust/Biomass, Mountaintop | High altitude station located on the dry, arid Tibet plateau in China. The site experiences clean or dusty air masses coming in from the west, and anthropogenically influenced and polluted air masses coming from the east (Kivekäs et al., 2009; Che et al., 2011) |

*All scattering instruments are TSI nephelometers

**Cape Point (CPT) had data loss issues in the 2012-2013 time period, so the period 2010-2011 was used instead





**Table 3** Number of hourly data points, plus median values and lower and upper quartiles for Scattering Ångström exponent and Absorption Ångström Exponent, Single Scattering Albedo, scattering coefficient ($\sigma_{sp}$), absorption coefficient ($\sigma_{ap}$) and inferred aerosol type at each monitoring station. All data are filtered by thresholds $\sigma_{sp} > 1$ Mm$^{-1}$ and $\sigma_{ap} > 0.5$ Mm$^{-1}$

| Station | # data points | SAE (lq, uq) | AAE (lq, uq) | SSA (lq, uq) | $\sigma_{sp}$ (Mm$^{-1}$) (lq, uq) | $\sigma_{ap}$ (Mm$^{-1}$) (lq, uq) | Aerosol type based on Cazorla et al. (2013) scheme | Aerosol type based on clustering of aerosol optical properties |
|---|---|---|---|---|---|---|---|---|
| ALT | 1648 | **1.27** (1.05, 1.43) | **0.86** (0.79, 0.95) | **0.93** (0.92, 0.94) | **9.69** (8.16, 12.11) | **0.75** (0.63, 0.90) | Large Coated Particles | Large coated (or processed) particles + EC/OC |
| AMY | 8914 | **1.57** (1.36, 1.75) | **1.22** (0.94, 1.42) | **0.92** (0.90, 0.95) | **107.72** (61.81, 189.54) | **8.72** (5.53, 13.44) | EC/OC | EC/OC |
| APP | 15547 | **2.11** (1.94, 2.26) | **1.20** (0.87, 1.48) | **0.92** (0.89, 0.94) | **24.46** (14.59, 38.17) | **2.13** (1.38, 3.19) | EC/OC | EC/OC |
| ARN | 8237 | **1.37** (0.97, 1.70) | **1.32** (1.16, 1.50) | **0.89** (0.85, 0.92) | **26.10** (16.7, 40.73) | **3.15** (1.83, 5.04) | EC/OC | EC/OC |
| BEO | 5775 | **1.87** (1.44, 2.07) | **1.31** (1.05, 1.55) | **0.92** (0.90, 0.94) | **22.64** (11.52, 40.04) | **1.94** (1.07, 3.21) | EC/OC | EC/OC |
| BND | 15257 | **2.01** (1.84, 2.17) | **1.15** (0.93, 1.34) | **0.93** (0.89, 0.95) | **33.06** (19.90, 55.14) | **2.69** (1.58, 4.17) | EC/OC | EC/OC |
| BRW | 2612 | **1.17** (0.78, 1.52) | **0.99** (0.89, 1.10) | **0.93** (0.90, 0.96) | **10.47** (7.87, 15.97) | **0.73** (0.60, 1.00) | Large Coated Particles | Large coated (or processed) particles + EC/OC |
| CPR | 5744 | **0.28** (0.17, 0.54) | **2.00** (1.16, 2.65) | **0.97** (0.96, 0.98) | **35.32** (24.33, 50.22) | **1.01** (0.71, 1.5) | Dust | Dust |
| CPT | 3158 | **0.67** (0.34, 1.14) | **1.12** (0.97, 1.31) | **0.96** (0.94, 0.97) | **21.31** (13.76, 29.79) | **1.14** (0.73, 2.45) | Dust/EC Mix | Mix |
| FKB | 5543 | **1.80** (1.59, 1.95) | **1.07** (0.98, 1.16) | **0.85** (0.79, 0.88) | **32.37** (18.12, 57.77) | **5.75** (3.17, 9.96) | EC/OC | EC/OC |
| GRW | 7960 | **-0.12** (-0.34, 0.19) | **0.62** (0.31, 0.85) | **0.97** (0.95, 0.98) | **30.73** (19.37, 47.42) | **0.84** (0.64, 1.29) | Large Coated Particles | Mix |
| GSN | 10731 | **1.51** (1.29, 1.70) | **1.21** (1.03, 1.34) | **0.93** (0.92, 0.95) | **61.85** (37.92, 106.47) | **4.59** (2.70, 7.40) | EC/OC | EC/OC |
| KPS | 8923 | **2.06** (1.90, 2.19) | **1.39** (1.24, 1.60) | **0.88** (0.85, 0.90) | **45.11** (25.27, 90.90) | **6.27** (3.61, 12.02) | EC/OC | EC/OC |
| LLN | 8294 | **1.94** (1.82, 2.08) | **1.11** (0.97, 1.25) | **0.91** (0.88, 0.93) | **24.02** (11.81, 40.00) | **2.39** (1.20, 4.56) | EC/OC | EC/OC |
| MLO | 2351 | **1.40** (0.85, 1.76) | **1.42** (1.08, 1.89) | **0.92** (0.85, 0.95) | **9.38** (4.88, 18.39) | **0.85** (0.64, 1.19) | Mix | Large coated (or processed) particles + EC/OC |
| NIM | 4527 | **0.32** (0.14, 0.64) | **1.66** (1.46, 1.22) | **0.91** (0.86, 0.94) | **91.02** (50.67, 185.24) | **9.25** (5.68, 16.05) | Dust | Dust |
| PGH | 4079 | **0.75** | **1.03** | **0.94** | **126.31** | **8.14** | Dust/EC Mix | Dust |





|  |  | (0.53, 0.92) | (0.88, 1.22) | (0.92, 0.95) | (66.48, 232.01) | (4.52, 126.31) |  |  |
|---|---|---|---|---|---|---|---|---|
| PVC | 4990 | **2.15** | **0.99** | **0.93** | **16.08** | **1.10** | EC/OC | EC/OC |
|  |  | (1.64, 2.50) | (0.68, 1.25) | (0.90, 0.95) | (10.19, 27.87) | (0.75, 1.82) |  |  |
| PYE | 481 | **0.98** | **0.50** | **0.98** | **40.00** | **0.69** | Large Coated Particles | Mix |
|  |  | (0.53, 1.29) | (0.30, 1.52) | (0.97, 0.99) | (26.59, 59.97) | (0.58, 1.00) |  |  |
| SGP | 14610 | **1.77** | **1.30** | **0.92** | **26.75** | **2.31** | EC/OC | EC/OC |
|  |  | (1.43, 2.06) | (1.05, 1.51) | (0.89, 0.94) | (16.06, 42.27) | (1.41, 3.42) |  |  |
| SPL | 8509 | **1.69** | **1.37** | **0.92** | **11.50** | **0.93** | EC/OC | Large coated (or processed) particles + EC/OC |
|  |  | (1.24, 2.03) | (1.22, 1.51) | (0.90, 0.94) | (7.79, 17.70) | (0.69, 1.35) |  |  |
| SUM | 462 | **1.93** | **1.04** | **0.93** | **8.06** | **0.64** | EC/OC | Large coated (or processed) particles + EC/OC |
|  |  | (1.62, 2.07) | (0.93, 1.16) | (0.91, 0.95) | (6.27, 11.58) | (0.55, 0.81) |  |  |
| THD | 5283 | **0.96** | **1.43** | **0.95** | **21.51** | **0.94** | Dust/EC Mix | Mix |
|  |  | (0.62, 1.43) | (1.14, 1.70) | (0.93, 0.97) | (13.09, 34.56) | (0.68, 1.4) |  |  |
| WLG | 6494 | **1.10** | **1.37** | **0.93** | **42.19** | **3.01** | Mix | Mix |
|  |  | (0.72, 1.35) | (1.22, 1.54) | (0.92, 0.95) | (20.08, 101.06) | (1.67, 6.16) |  |  |





**Table 4. Median AAE, SAE, SSA, and log($\sigma_{sp}$) values (along with corresponding interquartile spread) for each cluster resulting from the cluster analysis.**

| Cluster # | AAE | SAE | SSA | log($\sigma_{sp}$) | Sites included in cluster | Aerosol type according to Cazorla et al. (2013) matrix | Cluster commonality/ Site descriptions |
|---|---|---|---|---|---|---|---|
| 1 | 1.04 (0.99, 1.37) | 1.40 (1.27, 1.69) | 0.93 (0.92, 0.93) | 2.27 (2.23, 2.34) | ALT, BRW, MLO, SPL, SUM | Large coated particles + EC/OC | Remote Arctic or mountaintop with long-range transport aerosol or occasional local influence |
| 2 | 1.22 (1.21, 1.22) | 1.54 (1.53, 1.55) | 0.93 (0.92, 0.93) | 4.44 (4.29, 4.57) | AMY, GSN | EC/OC | Heavily polluted South Korean coastal sites |
| 3 | 1.20 (1.11, 1.31) | 1.94 (1.80, 2.06) | 0.92 (0.89, 0.92) | 3.26 (3.18, 3.48) | APP, ARN, BEO, BND, FKB, KPS, LLN, PVC, SGP | EC/OC | Primarily continental sites experiencing urban or biomass burning aerosol |
| 4 | 1.34 (1.19, 1.50) | 0.53 (0.43, 0.64) | 0.92 (0.92, 0.93) | 4.67 (4.59, 4.76) | NIM, PGH | Dust | Continental sites experiencing heavy dust loading and biomass burning aerosol |
| 5 | 2.00 | 0.28 | 0.97 | 3.56 | CPR | Dust | Coastal site experiencing occasional dust, biomass burning or pollution |
| 6 | 1.12 (0.62, 1.37) | 0.96 (0.67, 0.98) | 0.95 (0.94, 0.97) | 3.43 (3.07, 3.69) | CPT, GRW, PYE, THD, WLG | Mix | Coastal or remote sites experiencing occasional sea salt, dust, biomass burning or pollution aerosol |





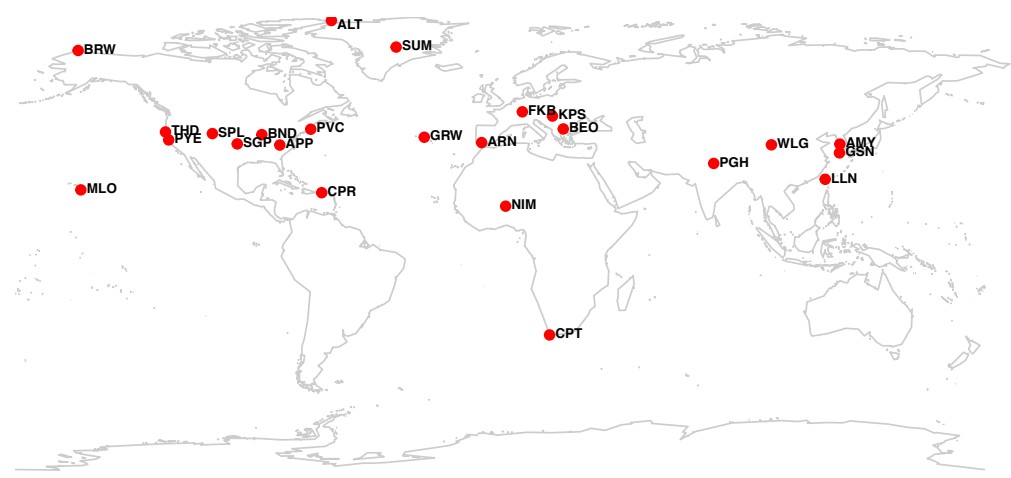

**Figure 1 Map of 24 in situ monitoring stations within the NOAA/ESRL Federated Aerosol Network that were utilized in this study. Locations are labeled with each site's 3-letter station abbreviation.**




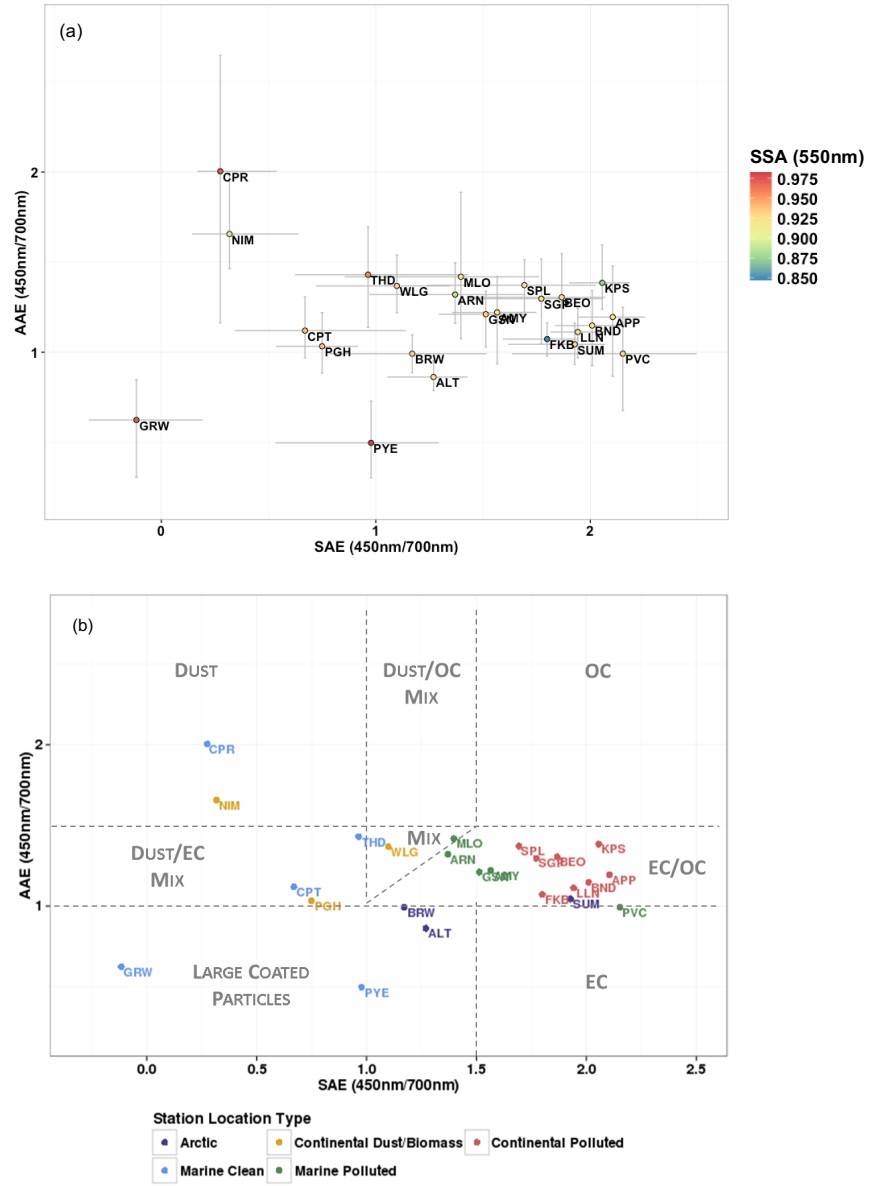

**Figure 2. AAE vs. SAE medians plotted for 24 in situ monitoring stations in the NOAA/ESRL federated network. (a) Bars represent interquartile values, and points are color coded by median SSA value at the station, (b) Points are color coded by station location type, and plot is overlaid with aerosol classification matrix from Cazorla et al. (2013)**





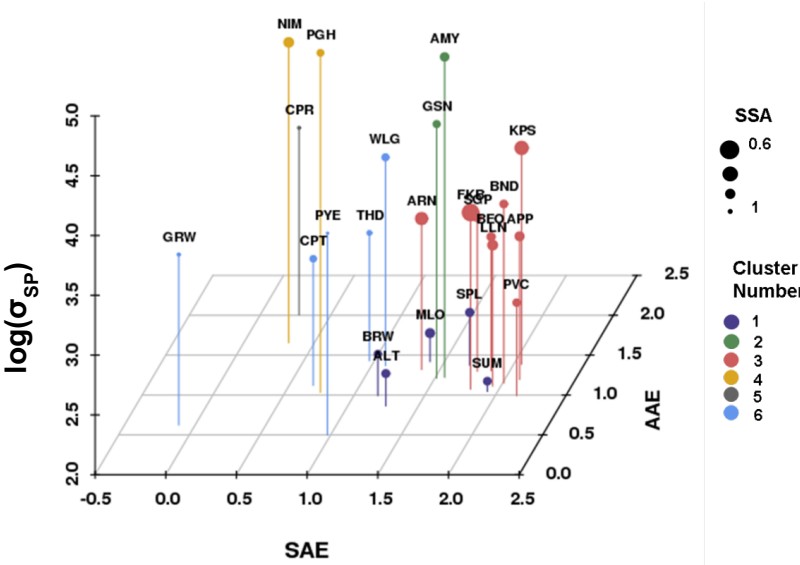

**Figure 3. 3D parameter space of SAE vs. AAE vs. log of scattering coefficient, $\sigma_{sp}$ (Mm$^{-1}$). Station points are colored by cluster number resulting from the clustering analysis, and sized by median SSA value.**

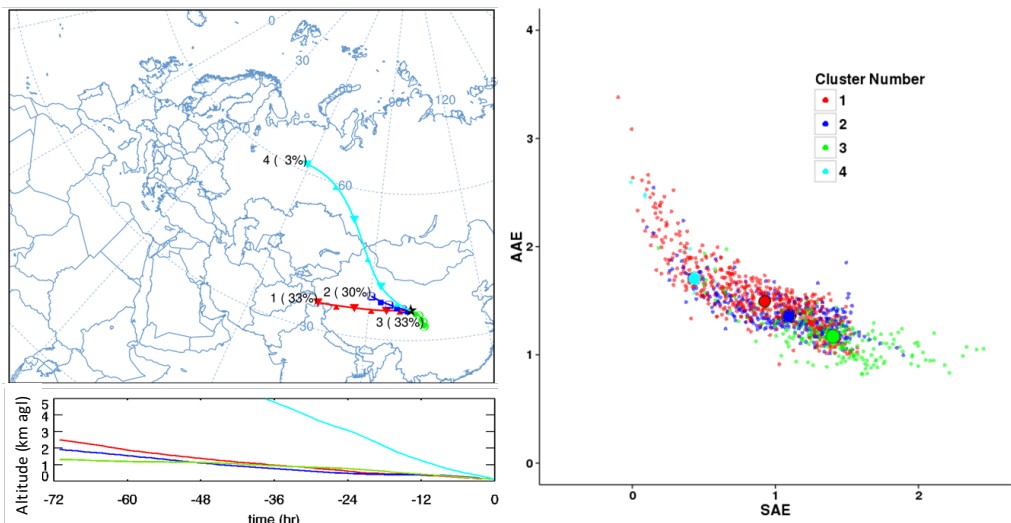

5    **Figure 4. Back trajectory map, back trajectory height (in kilometers above ground level) vs. time, and AAE vs. SAE plot space for Mt. Waliguan, China station WLG, all color coded by back trajectory cluster number. The percentage of air-mass back trajectories corresponding to each cluster are also shown next to the mean trajectories.**





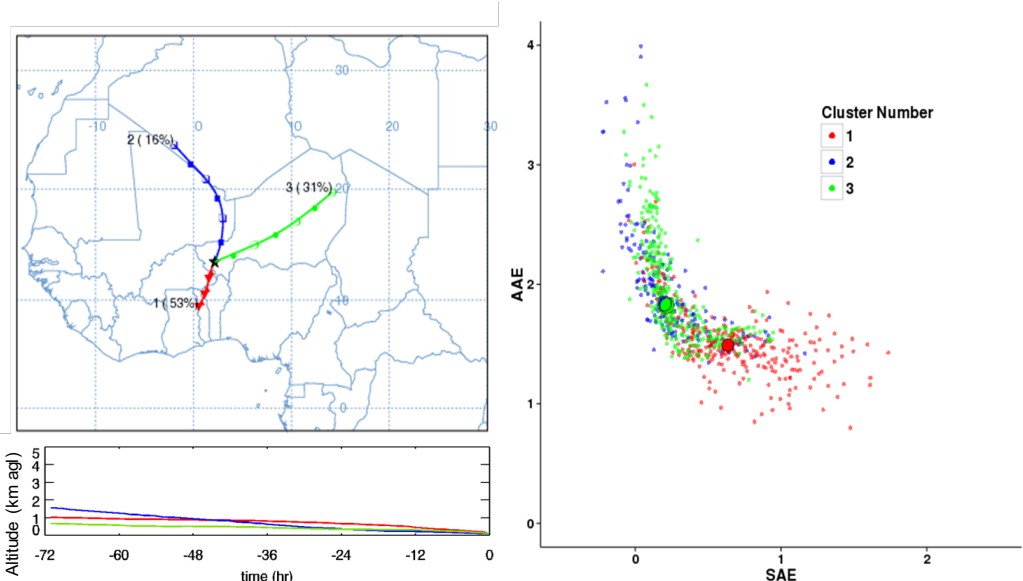

**Figure 5. Back trajectory map, back trajectory height (in kilometers above ground level) vs. time, and AAE vs. SAE plot space for Niamey, Niger station NIM, all color coded by back trajectory cluster number**

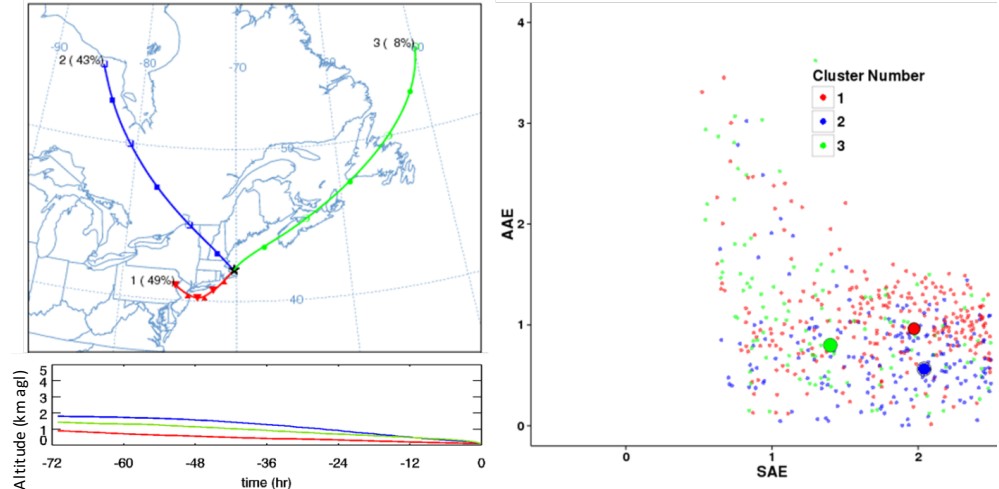

5    **Figure 6. Back trajectory map, back trajectory height (in kilometers above ground level) vs. time, and AAE vs. SAE plot space for Cape Cod, Massachusetts, USA station PVC, all color coded by back trajectory cluster number**

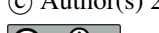



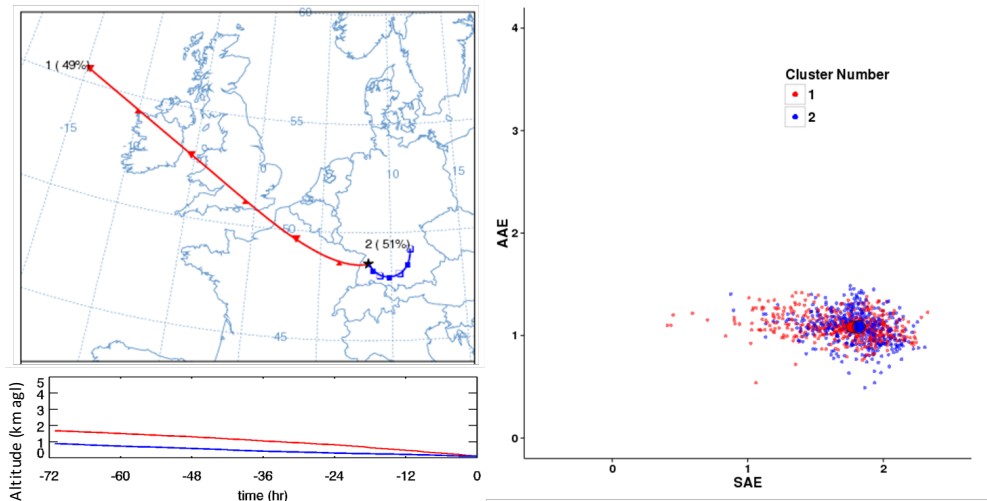

**Figure 7.** Back trajectory map, back trajectory height (in kilometers above ground level) vs. time, and AAE vs. SAE plot space for Black Forest, Germany station FKB, all color coded by back trajectory cluster number

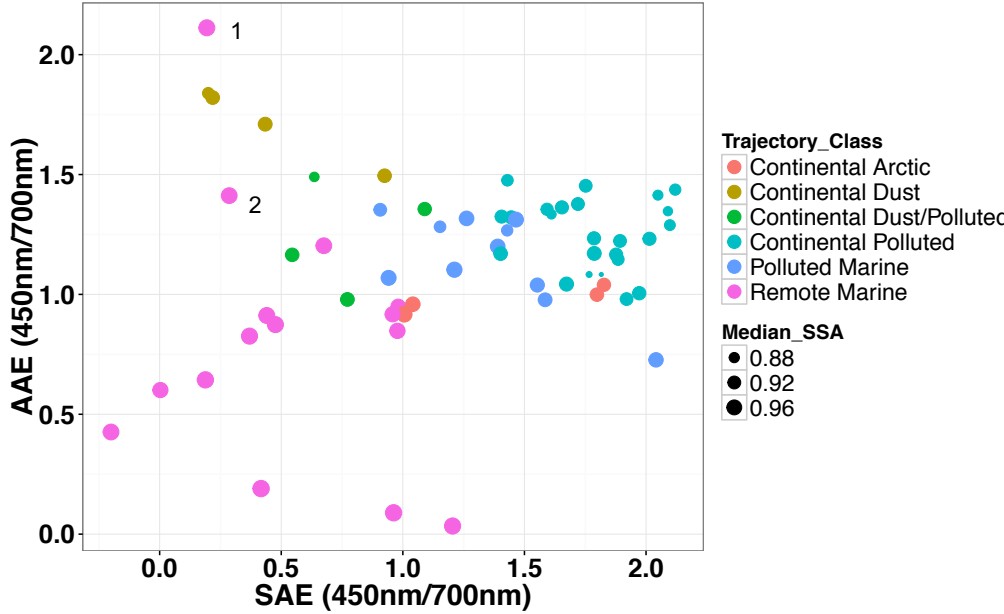

**Figure 8.** AAE vs. SAE medians plotted for all back trajectory clusters from 24 in situ monitoring stations in the NOAA/ESRL federated network. Points are colored by the trajectory classification, and sized by the median SSA value of measurements from that trajectory cluster, such that smaller points indicate low SSA values and larger point indicate high SSA value. The points labeled 1 and 2 are back trajectories from CPR that are outliers discussed in the text in Sect. 5.3