# Peer review of "Classifying aerosol type using in situ surface spectral aerosol optical properties"

_Atmospheric Chemistry and Physics, 2017_

## Referee Comment (RC1) · Anonymous Referee #1 · 2 May 2017

**GENERAL COMMENTS**

The paper addresses the important topic of aerosol type classification through aerosol spectral optical properties. Authors explore how existing schemes reported in the literature may be applied and possibly improved. The paper demonstrates that there still are several uncertainties in all classification schemes proposed here, and discusses major ambiguities and limitations. No significant improvement is, however, proposed.

The topic is fully within the scope of ACP, and I recommend this paper to be published in ACP after the following issues are addressed.

**SPECIFIC COMMENTS**

1. The meaning of AAE as indicator of aerosol properties still contains uncertainties that have not been formally assessed in the scientific community:

(i) AAE=1 as indicator of BC.

Although AAE =1 has often been attributed to BC, several studies have demonstrated that AAE depends on both aerosol composition (and in particular, to the BC-to-OA ratio more than to the BC mass concentration alone), and size (e.g., Lack and Langridge, 2013; Saleh et al., 2014; Costabile et al., 2017). Caution should therefore be taken in interpreting AAE=1 as indicator of BC only. As an example, if AAE=1 indicates BC only, cluster 1 will indicate BC (pag.9 lines 35-38); however, authors ackowledge that "the generally low scattering coefficients at these sites indicate the presence of anthropogenic aerosols is infrequent" (pag.10, lines 6-8).

(ii) AAE>2 as indicator of brown carbon.

Although several studies have indicated that AAE values from 2 to 6 may indicate brown carbon (e.g., Andreae and Gelencsér, 2006; Moosmüller et al., 2011), authors do not consider brown carbon at all in this paper. In fact, the contribution of BrC aerosol with large AAE values (2-6) may be less significant at the regional remote/rural sites analysed here, as discussed at pag.18, line 17. Therefore, the paper should mention that the proposed analysis may fail at urban sites and more polluted sites. Indeed, authors ackowledge (pag.8, line 28) that Cazorla et al.'s scheme may fail for sites in proximity to aerosol sources.

(iii) AAE of carbonaceous aerosols.

The paper mentions that carbonaceous aerosols have AAE  $\sim 1.2$  (e.g., pag.10, line 10). However, carbonaceous aerosols include both BC, OA and BrC, and their relative contribution in particle with different size can cause a larger variability in AAE values.

**2. Overall presentation structure.**

Results section (Sect.5) is too long and a little bit confusing. In particular:

- cluster analysis results (pag. 9,10,11) are somehow ambiguous, as authors ack-owledge (pag.11, lines 20-24);

- description of multivariate cluster analysis (pag.9, lines 1-25) should be moved to a separate paragraph (e.g., in a data analysis section).

Discussion section (Sect.6) is the most interesting part of the paper, but it should be better clarified in some parts. In particular, I recommend to address:

(i) Uncertainties in the AAE attribution method mentioned above;

(ii) Differences between columnar and in situ surface spectral optical properties.

Authors apply thresholds from Cazorla et al.'s scheme - obtained by columnar measurements (AERONET) - to interpret in situ surface data. Although results may be consistent, in principle this is not completely correct. In fact, aerosol spectral optical properties measured by columnar and in-situ ground instruments may in principle differ significantly. Relevant thresholds seem, however, to be consistent in Tab.1. Please, add a more comprehensive discussion (this is only sketched in the discussion section, pag. 18, lines 1-10).

(iii) Classification of measurement site.

Authors demonstrate that none of the classification schemes applied here can classify measurement sites. None of the sites has indeed only a single dominant aerosol type (pag.12, lines 6-8). Only continental polluted sites are well classified. This is a reasonable conclusion, as these classification schemes should classify aerosol type, not measurement site. Please, discuss this point more clearly (this is roughly discussed in Sect.6).

**СЗ**

3. Sampling line description is not clear enough:

- Are all sites but SUM equipped with PM10 sampling heads?

- Heated inlets might cause losses in organic aerosol and volatile compounds. This can influence aerosol spectral optical properties. Is heating performed at all sites?

Please, add more details.

**TECHNICAL CORRECTIONS**

- Pag.8, line 2: is
- Tab.4 : check log ( $\sigma$ )
- Pag.12, line 20: "is"

**REFERENCES**

Andreae, M. O. and Gelencsér, A.: Black carbon or brown carbon? The nature of light-absorbing carbonaceous aerosols, Atmos. Chem. Phys., 6, 3131-3148, doi:10.5194/acp-6-3131-2006, 2006.

Costabile, F., Gilardoni, S., Barnaba, F., Di Ianni, A., Di Liberto, L., Dionisi, D., Manigrasso, M., Paglione, M., Poluzzi, V., Rinaldi, M., Facchini, M. C., and Gobbi, G. P.: Characteristics of brown carbon in the urban Po Valley atmosphere, Atmos. Chem. Phys., 17, 313-326, doi:10.5194/acp-17-313-2017, 2017.

Lack, D. A. and Langridge, J. M.: On the attribution of black and brown carbon light absorption using the Ångström exponent, Atmos. Chem. Phys., 13, 10535-10543, doi:10.5194/acp-13-10535-2013, 2013.

Moosmüller, H., Chakrabarty, R. K., Ehlers, K. M., and Arnott, W. P.: Absorption Ångström coefficient, brown carbon, and aerosols: basic concepts, bulk matter, and spherical particles, Atmos. Chem. Phys., 11, 1217-1225, doi:10.5194/acp-11-1217-2011, 2011.

Saleh, R., Robinson, E. S., Tkacik, D. S., Ahern, A. T., Liu, S., Aiken, A. C., Sullivan, R. C., Presto, A. A., Dubey, M. K., Yokelson, R. J., Donahue, N. M., and Robinson, A. L.: Brownness of organics in aerosols from biomass burning linked to their black carbon content, Nat. Geosci., 7, 647-650, 2014.

**C5**

---

## Author Comment (AC1) · 12 Jul 2017

**Response to Anonymous Referee #1**

Thank you, anonymous referee #1, for your thorough and constructive comments. The manuscript is much improved after your input.

Our response is structured as follows: original comments from reviewer #1 are bolded, our responses are in italics, and the revised portions of the manuscript follow in quotation marks with specific changes/additions in red.

General Comments
**The paper addresses the important topic of aerosol type classification through aerosol spectral optical properties. Authors explore how existing schemes reported in the literature may be applied and possibly improved. The paper demonstrates that there still are several uncertainties in all classification schemes proposed here, and discusses major ambiguities and limitations. No significant improvement is, however, proposed. The topic is fully within the scope of ACP, and I recommend this paper to be published in ACP after the following issues are addressed.**

Specific Comments
   *1.* **The meaning of AAE as indicator of aerosol properties still contains uncertainties that have not been formally assessed in the scientific community.**
      i. **AAE = 1 as indicator of BC. Although AAE = 1 has often been attributed to BC, several studies have demonstrated that AAE depends on both aerosol composition (and in particular, to the BC-to-OA ratio more than to the BC mass concentration alone), and size (e.g., Lack and Langridge, 2013; Saleh et al., 2014; Costabile et al., 2017). Caution should therefore be taken in interpreting AAE=1 as indicator of BC only. As an example, if AAE = 1 indicates BC only, cluster 1 will indicate BC (pag. 9 lines 35-38); however, authors acknowledge that "the generally low scattering coefficients at these sites indicate the presence of anthropogenic aerosols is infrequent" (pag. 10, lines 6-8).**

      *Thank you for the comment. It is certainly important to mention the uncertainties associated with the AAE =1 indicating black carbon, and an expanded discussion of AAE uncertainties as related to BC has been added to the manuscript, on Pag. 2 and in the discussion sect (new Sect. 7 – see author response to reviewer comment 2(i) below).*

      Pag. 2, line 34-Pag. 3, line 2: "Black carbon (BC), for example, has a theoretical AAE value around 1, while dust aerosol typically has AAE values greater than 2 (Bergstrom et al., 2002, 2007; Kirchstetter et al., 2004), though AAE of ambient aerosol is likely to evolve with atmospheric processing and will depend strongly on composition (e.g., black carbon-to-organic aerosol ratio), coating and size (Saleh et al., 2014; Costabile et al., 2017; Moosmüller et al., 2011)."

      ii. **AAE > 2 as indicator of brown carbon. Although several studies have indicated that AAE values from 2 to 6 may indicate brown carbon (e.g., Andreae and Gelencsér, 2006; Moosmüller et al., 2011), authors do not consider brown carbon at all in this paper. In fact, the contribution of BrC aerosol with large AAE values (2-6) may be less significant at the regional remote/rural sites analysed here, as discussed at pag. 18, line 17. Therefore, the paper should mention that the proposed analysis may fail**

**at urban sites and more polluted sites. Indeed, authors acknowledge (pag. 8, line 28) that Cazorla et al.'s scheme may fail for sites in proximity to aerosol sources.**

*We agree that brown carbon should not be neglected, and did not intend to do so. The authors implicitly assumed that brown carbon was encompassed by the 'OC' designation from the Cazorla et al. (2013) matrix (Cappa et al., 2016 also made this assumption). However, we realize we should specify that BrC is the light-absorbing portion of OC. We have changed the aerosol classification matrix to that of Cappa et al. (2016), which includes BrC (in other words, light-absorbing OC). This change is reflected in Fig. 2(b) and in the text of the results section.*

*Nonetheless, none of the sites in this analysis show AAE values greater than 2, so this does not affect the results of the analysis. CPR does have a median AAE value of 2, but the supporting evidence is strong that this is likely dust from long range transport, mixed with BrC (see Pag. 11, lines 11-20, and references Prospero et al., 2014; Denjean et al., 2016; Kalashnikova and Kahn, 2008; and Reid et al., 2003). There is a possibility that BrC is not identified by AAE values greater than 2 due to limitations from the instruments measuring aerosol light absorption. Since the AAE values derived from PSAP and CLAP measurements for this analysis are for the 450/700nm wavelength pair, and since BrC absorbs most strongly in UV (which is not measured by the PSAP and CLAP), it is possible that this method would not positively identify BrC even if it was present at the measurement sites. There is no way for us to confirm this given the data that are available to us, but it is a potential limitation of the data that is worth mentioning.*

*It is not clear why reviewer #1 suggests that the proposed analysis may fail at urban sites and more polluted sites. In fact, there are many urban and/or polluted sites that are included in this analysis for which the classification scheme works well (e.g., GSN, AMY, FKB).*
*Even polluted sites dominated by BrC would be well classified with the existing schemes, as they would specifically identify OC/BrC when AAE > 2, assuming the potential instrumental limitations mentioned above are not an issue.*

iii. **AAE of carbonaceous aerosols. The paper mentions that carbonaceous aerosols have AAE ~ 1.2 (e.g., pag. 10, line 10). However, carbonaceous aerosols include both BC, OA, and BrC, and their relative contribution in particle with different size can cause a large variability in AAE values.**

*More careful use of the term 'carbonaceous aerosol' has been used throughout, and the term has been replaced by more specific reference to BC, OC or BrC as needed. The authors agree that in the example on previous version pag. 10, line 10, the term 'carbonaceous aerosol' was misused in conjunction with the designation of AAE~1.2. This has now been changed, see below.*

Now in supplemental materials, page 2: "Cluster 2 includes AMY and GSN, the two coastal stations located in South Korea, and is characterized by high aerosol loadings (high $\sigma_{sp}$), small aerosol particles (SAE~1.5) and BC dominated aerosols (AAE~1.2)."

**2. Overall presentation and structure**
**Results section (Sect. 5) is too long and a little bit confusing. In particular:**

i.   **Cluster analysis results (pag. 9, 10, 11) are somehow ambiguous, as authors acknowledge (pag. 11, lines 20-24)**

*Discussion of individual cluster results from the multivariate cluster analysis has been moved from the manuscript to supplemental materials in order to shorten the results section and highlight the important takeaways from the analysis. Although the results of the cluster analysis are indeed somewhat ambiguous, this is one of the important results from this aerosol classification technique. Namely, a cluster analysis does not eliminate uncertainties in aerosol classification schemes, though it does help give a more complete picture of aerosol conditions at the measurement sites. This is reiterated in the discussion section.*

Pag. 17, lines 4-8: "An anticipated advantage to the multivariate cluster analysis was that it would help to reduce ambiguity in results of aerosol typing schemes, though this was not the case with every cluster. Rather than falling more surely within the optical property thresholds of one aerosol type, the median optical properties of a few clusters still fell on the cusp of two or more aerosol type thresholds. This left the aerosol type of some clusters uncertain, particularly for clusters with coastal and/or remote sites."

ii.   **Description of multivariate cluster analysis (pag. 9, lines 1-25) should be moved to a separate paragraph (e.g., in a data analysis section).**

*A new data analysis section has been added (Sect. 5), and description of data analysis methods for each of the classification techniques has been added and/or moved to that section to help shorten the results section (now Sect. 6). See Sect. 5 in newest version of manuscript.*

**Discussion section (Sect. 6) is the most interesting part of the paper, but it should be better clarified in some parts. In particular, I recommend to address:**
i.   **Uncertainties in the AAE attribution method mentioned above**

*A paragraph expanding on uncertainties in the AAE attribution method (particularly for BC) has been added to the discussion section (now Sect. 7).*

Pag. 17, line 32- Pag. 18, line 2: "It should be mentioned that the success of aerosol classification schemes is largely dependent on uncertainties in AAE attribution. The scientific community has yet to fully assess AAE as an indicator of aerosol composition. Although AAE=1 is often taken to indicate black carbon, some studies show that this largely depends on aerosol composition and size, as well as the age of the particle and atmospheric processing that it endures (Lack and Langridge, 2013; Saleh et al., 2014; Costabile et al., 2017; Moosmüller et al., 2011). Furthermore, the accuracy of these aerosol classification methods are only as accurate as the AAE value is an indication of the aerosol composition. As the scientific community advances our understanding of AAE and its relationship to aerosol composition and size, these aerosol classification schemes should be refined."

ii.   **Differences between columnar and in situ surface spectral optical properties. Authors apply thresholds from Cazorla et al.'s scheme – obtained by columnar measurements (AERONET) – to interpret in situ surface data. Although results may**

**be consistent, in principle that is not completely correct. In fact, aerosol spectral optical properties measured by columnar and in-situ ground instruments may in principle differ significantly. Relevant thresholds seem, however, to be consistent in Tab. 1. Please, add a more comprehensive discussion (this is only sketched in the discussion section, pag. 18, lines 1-10).**

*It is true that aerosol spectral optical properties measured by columnar and in-situ ground instruments may in principle differ. Since the Cazorla et al. (2013) aerosol classification matrix was based on ambient AERONET measurements, it could include confounding effects of water uptake by aerosols (depending on the aerosol hygroscopicity). Without knowing the ambient RH of the environment during the Cazorla et al. (2013) measurements, we have no way of knowing how the matrix would differ if the data was unaffected by potential water uptake. However, the matrix is just a guiding classification scheme- not a definitive answer for aerosol type. Furthermore, the classification matrix still seems reasonable given what we know about aerosol optical properties, and it does seem to work reasonably well for our data. Moreover, Cappa et al. (2016) use in-situ data to retool the Cazorla et al. (2013) classification matrix, and though their categories are slightly more detailed, the big picture categories align very well with the Cazorla et al. (2013) classifications.*

*In order to address the concerns of the reviewer that we used a matrix developed with AERONET columnar measurements and applied it to in-situ data, we have changed the matrix to that of Cappa et al. (2016), which was developed based on the Cazorla et al. (2013) matrix, but modified using their results from in situ data. This does not substantially change any results from our analysis, since the matrices are very similar (see comparison of the matrices and added text describing the similarities below).*

*Cazorla et al. (2013) matrix is on the left side, Cappa et al. (2016) matrix is on the right.*

[Figure]

[Figure]

Pag. 8, lines 13-23: "It should be noted that the Cappa et al. (2016) and Cazorla et al. (2013) matrices are very similar. Both designate high SAE and high AAE values as BrC or mixed BC/BrC (though Cazorla et al. (2013) refers to BrC as OC). Both designate low SAE values and high AAE values as dust or dust mixed with BC and BrC, and both suggest an AAE value around 1, accompanied by higher SAE values indicates aerosol populations dominated by BC. Three main differences between the matrices can be identified. The Cappa et al. (2016) matrix makes more specific designations of aerosol

mixtures (e.g., adds 'mixed dust/BC/BrC' and 'large particle/BC mix'). The Cappa et al. (2016) matrix also replaces the Cazorla et al. (2013) matrix designation of 'large coated particles' with 'large particle/low absorption mix or large black particles'. Finally, the Cappa et al. (2016) matrix replaces the Cazorla et al. (2013) matrix designation of 'EC' with 'small particle/low absorption mix'. We chose to primarily use the Cappa et al. (2016) matrix since it is based on in situ data (Cazorla et al. (2013) is based on AERONET data), and since the aerosol designations seemed to align most closely with our data. Results are presented in Sect. 6.1."

iii. **Classification of measurement site. Authors demonstrate that none of the classification schemes applied here can classify measurement sites. None of the sites has indeed only a single dominant aerosol type (pag. 12, lines 6-8). Only continental polluted sites are well classified. This is a reasonable conclusion, as these classification schemes should classify aerosol type, not measurement site. Please, discuss this point more clearly (this is roughly discussed in Sect. 6).**

*The classification schemes applied here are not meant to classify measurement sites. Rather, the use of measurement site classifications here is meant to help validate aerosol type inferred from the aerosol classification techniques, since chemical measurements are not available at these sites to validate aerosol types. This is an imperfect method- not only because sites may have more than one dominant aerosol (e.g., a polluted marine site that measures both anthropogenic pollution aerosols and natural sea salt aerosols), but also because sites may measure aerosols that are unexpected given the site classification due, for example, to long range transport (e.g., a clean marine site measuring pollution aerosol that was transported a long distance). However, given the remaining ambiguities, the site classifications paired with previous analyses of these sites and the aerosols they measure do provide useful information that shows when/where the aerosol classification techniques are most useful, and when/where they tend to fail.*

3. **Sampling line description is not clear enough**
   i. **Are all sites but SUM equipped with PM$_{10}$ sampling heads?**

   *All measurements analyzed here are collected from 10 µm size cuts, with the exception of those from SUM, which utilizes a 2.5 µm inlet (see Pag. 7, lines 17-19).*

   ii. **Heated inlets might cause losses in organic aerosol and volatile compounds. This can influence aerosol spectral optical properties. Is heating performed at all sites? Please, add more details.**

   *Thank you for encouraging us to add this information to the manuscript. The following paragraph has been added:*

   Pag. 6, lines 4-11: "In order to minimize aerosol hygroscopic effects, measurements at all stations (except SUM and SPL) are made at a reduced relative humidity (RH < 40%) by heating the inlet air or by diluting with filtered, dry air. The inlets at most sites are either gently heated (heating does not exceed 40°C) with a stack heater or a small heater by the impactor, and are only utilized if the relative humidity exceeds 40%. Although heating the sampling inlet can cause loss of organic and volatile aerosol material, which can alter the aerosol spectral optical properties, this is not expected to substantially impact results

here. Studies that analyze the amount of volatile components removed at 40°C (by a thermal denuder) is less than 10% (Mendes et al., 2016; Huffman et al., 2009). For this particular study, we do not have the data necessary to evaluate the extent to which aerosol optical properties are affected by the heating, but evidence from other studies suggests the effect is likely small."

**Technical Corrections**

**-Pag. 8, line 2: is**

*This suggested correction was not made since we intended to write 'data are' (data is a plural of datum).*

**-Tab. 4: check log(sigma)**

*Thank you for catching this. We changed Iog($\sigma_{sp}$) to log($\sigma_{sp}$) in Tab. 4.*

**-Pag. 12, line 20: "is"**

*We could not identify what needed to be changed to 'is' on Pag. 12, line 20 in the old version of the manuscript.*

**References**

Andreae, M.O., and Gelencsér, A.: Black carbon or brown carbon? The nature of light-absorbing carbonaceous aerosol, Amtos. Chem. Phys., 6, 3131-3148, doi: 10.5194/acep-6-3131-2006, 2006.

Costabile, F., Gilardoni, S., Barnaba, F., Di Ianni, A., Di Liberto, L., Dionisi, D., Manigrasso, M., Paglione, M., Poluzzi, V., Rinaldi, M., Facchini, M.C., and Gobbi, G.P.: Characteristics of brown carbon in the urban Po Valley atmosphere, Atmos. Chem. Phys., 17, 313-326, doi: 10.5194/acep-17-313-2017, 2017.

Lack, D.A. and Langridge, J.M.: On the attribution of black and brown carbon light absorption using the Angstrom exponent, Atmos. Chem. Phys., 13, 10535-10543, doi: 10.5194/acp-13-10535-2013, 2013.

Moosmuller, H. Chakrabarty, R.K., Ehlers, K.M., and Arnott, W.P.: Absorption Angstrom coefficient, brown carbon, and aerosols: basic concepts, bulk matter, and spherical particles, Atmos. Chem. Phys., 11, 1217-1225, doi:10.5194/acp-11-1217-2011, 2011.

Saleh, R., Robinson, E.S., Tkacik, D.S., Ahern, A.T., Liu, S., Aiken, A.C., Sullivan, R.C., Presto, A.A., Dubey, M., Yokelson, R.J., Donahue, N.M., and Robinson, A.L.: Brownness of organics in aerosols from biomass burning linked to their black carbon content, Nat. Geosci., 7, 647-650, 2014.

*Thank you for the useful references. We have incorporated these into the manuscript as needed, and have added them into the manuscript reference list.*

---

## Author Comment (AC2) · 12 Jul 2017

**Response to Anonymous Referee #3**

The authors thank anonymous referee #3 for their useful comments. The manuscript is better after considering referee #3's suggestions.

Our response is structured as follows: original comments from reviewer #1 are bolded, the author's responses are in italics, and the revised portions of the manuscript follow in quotation marks and red letters.

**General Comments**
**The authors present an important topic and explore a variety of aerosol classification schemes using an extensive data set with different aerosol types. This analysis allows authors to discuss about the differences between classification schemes and one of the main conclusions is that all classification schemes fail at some point, and some aerosol types are missing. This is, there is no combination of extensive and/or intensive optical properties that allow the perfect classification of aerosol types and it looks like knowing the measurement sites is needed, or helps discriminate the aerosol type. This should be included in the conclusions. Also the conclusions should explicitly indicate the specific ideas for future work needed (last paragraph on conclusion section).**

**The paper is within the scope of ACP and presents an interesting analysis that show the goodness and weakness of the different aerosol classification schemes presented and propose other approaches for the classification of aerosol. I recommend the publication of this paper.**

*Thank you for the comment; it is a good suggestion to bolster the conclusion by stating specifically that there is no 'silver bullet' classification scheme that can deduce aerosol type from only a combination of intensive and extensive aerosol optical properties. The authors also added specific ideas for future work needed into the conclusion. Please see changes to manuscript below.*

Pag. 20, lines 9-17: "This study has further assessed existing aerosol typing schemes, provided additional methods that can be implemented to reduce ambiguity in typing schemes, elucidate aerosol conditions that accompany aerosol type, and allow for identification of multiple aerosol types at one site. A major conclusion from the analysis, however, is that there is no combination of extensive and/or intensive optical properties that allow for a perfect classification of aerosol types. Prior knowledge of the measurement site can help inform aerosol classification schemes, but obscurity remains in these techniques. Furthermore, this paper highlighted the need for further analyses and suggests specific ideas for future work needed to progress and refine aerosol typing schemes that infer aerosol type from optical properties. Namely, repeating this analysis with concurrent aerosol chemical and optical measurements to verify aerosol classification thresholds will be essential to expand and improve aerosol classification schemes."

---

## Author Response (AR2)

**Author's Response**

Dear Dr. Petzold,

Thank you for the suggested minor revisions. Each has been addressed below, including (1) An added line break after 'particles' in the bottom right corner of Figure 2, (2) Enhanced resolution of Figures 4-7 including grid lines (as a substitute for minor grid marks) and increased axis font size, (3) Changed 'EC mixtures' to 'BC' mixtures on page 16, line 25.

You will notice slight difference between the new and old Figure 6, as a small error having to do with merging data with different time zones was found in the analysis code for PVC. This has been corrected, and the analysis text has been changed slightly to reflect the change; however, this does not at all change the overall results or conclusions of the paper. This error was long ago found and already corrected for Figure 8.

Kind Regards,

Lauren Schmeisser & co-authors

[revised manuscript text omitted]